# DIVIDE AND ORTHOGONALIZE: EFFICIENT CONTINUAL LEARNING WITH LOCAL MODEL SPACE PROJECTION

## ABSTRACT

Continual learning (CL) has attracted increasing interests in recent years due to the need for a learning model to continuously learn new tasks without forgetting the previously learned knowledge. However, existing CL methods require either an extensive amount of resources for computing gradient projections or storing a large number of old tasks' data. These limitations necessitate low-complexity CL algorithmic design. In this paper, we propose a local model space projection (LMSP) based efficient continual learning framework, which not only significantly reduces the complexity of computation, but also enables forward and backward knowledge transfer. We also theoretically establish the convergence performance of the proposed LMSP approach. Extensive experiments on several public datasets demonstrate the efficiency of our approach.

## 1 INTRODUCTION

Humans have the unique ability to continuously learn new tasks throughout their lives without forgetting their previously learned knowledge. This impressive capability has recently inspired the efforts in the machine learning community to develop similar capabilities for deep neural network (DNN)-based machine learning models, which is termed continual learning (CL). However, one of the most significant challenges in CL is that DNN models are known to suffer from the problem of "catastrophic forgetting", i.e., the performances of the learnt old tasks decay after learning new tasks. In the literature, numerous strategies have been proposed to address the catastrophic forgetting challenge in CL. Existing forgetting mitigation approaches can be classified into three major categories: i) experience replay, ii) regularization, and iii) orthogonal projection (see Section 2 for more in-depth discussions). Generally speaking, both experience-replay- and regularization-based methods require access to old tasks' data during the learning of a new task, which is infeasible in cases where old tasks' data leave the system once their learning is finished. In contrast, orthogonal-projection-based methods update the model in the direction orthogonal to the subspace of old tasks, which does *not* require the access to old tasks' data – a highly desirable feature for CL in practice.

We note, however, that due to a number of technical challenges, developing practical orthogonal-projection-based CL approaches remains highly non-trivial. The first major challenge of orthogonal-projection-based CL approaches stems from the projection operation, which typically relies on singular-value decomposition (SVD)(Lin et al., 2022a;b). These methods perform SVD layer-wise SVD after the training of each task. It is well-known that the SVD operation costs $O(n^3)$ complexity for a $n$-dimensional model, which grows rapidly as $n$ increases. With the ever-increasing widths and depths of large and deep learning models, computing such layer-wise SVDs upon the completion of each new task's training also becomes more and more difficult.

Another key challenge of the standard orthogonal-projection-based CL approaches lies in the inherent difficulty in facilitating *forward and backward knowledge transfer* (i.e., the learning of new tasks benefiting from the acquired knowledge from old tasks, and the knowledge learnt from new tasks further improves the performance of old tasks), when new task has strong similarity with some old tasks. To date, it remains unclear how to design *computation-efficient* orthogonal-projection-based CL methods without forgetting while enjoying forward-backward knowledge transfers. This motivates us to pursue a new efficient orthogonal-projection-based CL design to fill this gap in the CL literature.

The main contribution of this paper is that we propose an efficient orthogonal-projection-based CL method based on local model space projection (LMSP), which not only signficantly reduces the

complexity of SVD basis computation, but also facilitates forward and backward knowledge transfers without sacrificing too much performance. The main results and technical contributions of this paper are as follows:

- Our proposed LMSP-based orthogonal projection approach is based on the basic idea of "divide and orthogonalize" principle, where we approximate the per-layer parameter matrix by a set of local low-rank matrices defined by a set of anchor points, which significantly reduces the computational complexity from $\mathcal{O}(n^3)$ to $\mathcal{O}(n^2)$ in performing projections with a minor projection error.

- We theoretically show that our proposed LMSP-based orthogonal projection approach achieves an $\mathcal{O}(1/K)$ convergence rate performance under both convex and non-convex settings, where $K$ is the number of iterations. Moreover, we further prove the forward and backward knowledge transfers of the proposed LMSP-based orthogonal projection approach.

- Based on extensive experiments, we show that our proposed LMSP-based orthogonal projection approach achieves comparable results to those of all state-of-the-art baselines on four public datasets in terms of both training accuracy and forward/backward knowledge transfer, while not sacrificing too much performance. We further conduct ablation studies to verify the efficiency and effectiveness of each key component in our LMSP-based algorithmic design.

## 2 RELATED WORK

In this section, we provide a quick overview on continual learning and local low-rank model approximation to further motivate this research and put our work in comparative perspectives.

**1) Continual Learning:** Continual learning (CL), also known as lifelong learning and incremental learning, is an emerging area in machine learning research that has attracted a significant amount of interests recently. CL addresses the challenge of enabling a machine learning model to accumulate knowledge and adapt to new tasks that arrive sequentially over time Chen & Liu (2017). A key goal of CL is to avoid "catastrophic forgetting" (McCloskey & Cohen, 1989; Abraham & Robins, 2005), i.e., a model's performance on previously learned tasks decays upon learning new tasks. To mitigate catastrophic forgetting in CL, various methodologies and strategies have been proposed:

- *Regularization-Based Approaches:* Regularization approaches use regularization prevent a learning model from overfitting to new data. For example, elastic weight consolidation (EWC) Kirkpatrick et al. (2017a) regularizes the updates on weights based on their significance for previous tasks using the Fisher information matrix. Aljundi et al. (2018) uses an unsupervised and online way to evaluate the model outputs sensitivity to the inputs and penalizes changes to important parameters.

- *Replay-Based Approaches:* Replay-based approaches store and replay old tasks' data to help models retain knowledge. For example, generative replay Shin et al. (2017) generates data samples from past tasks. In experience replay Chaudhry et al. (2019b), a model replays previous experiences in a controlled manner. Techniques such as experience replay with replay buffer (ER-RB) Lillicrap et al. (2019) and generative adversarial networks (GANs) Goodfellow et al. (2020) have also been developed to enhance the efficiency of these mechanisms.

- *Orthogonal-Projection-Based Approaches:* In regularization- and replay-based approaches, a major limitation is that the learner needs to have access to data of the old tasks. However, this requirement could be cumbersome or even infeasible due to data privacy and other restrictions in practice. To address this challenge, researchers have proposed to learn the the new tasks and update the model in the orthogonal subspace of the old tasks Chaudhry et al. (2020), which alleviates the needs for accessing old tasks' data. State-of-the-art orthogonal-projection-based approaches, include, e.g., Lin et al. (2022a), where the correlations between old and new tasks.

Due to the salient features of the orthogonal-projection-based approaches, we focus on the orthogonal-projection-based approach in this paper. However, a key challenge of the orthogonal-projection-based CL approach is that the computation of orthogonal subspace is highly expensive as the model size gets large. This motivates us to propose a local model space projection (LSMP) in this paper.

**2) Local Model Approximation:** Low-rank approximation (LRA) techniques have been widely applied in the areas of matrix factorization (Billsus & Pazzani, 1998; Mnih & Salakhutdinov, 2007; Salakhutdinov & Mnih, 2008; Candes & Plan, 2009). The basic idea of these existing works is to

represent a given matrix by a product of lower-rank matrices that capture the essential structure of the original matrix. Local low-rank approximation (LLRA) extends LRA to preserve low-rank structures in localized regions of matrices. LLRA has been applied in various applications, such as recommendation Beutel et al. (2017); Sarwar et al. (2002); Christakopoulou & Karypis (2018), collaborative filtering George & Merugu (2005); Lee et al. (2014); Koren (2008). For example, Lee et al. (2013) proposed a local low-rank matrix approximation (LLORMA) method, which finds anchor points of the matrix and estimates local low-rank matrices in the neighborhood surrounding each anchor point. Then, a weighted sum of the local matrices is used to approximate the original matrix, where the weight is the similarity between the pair of anchor points. Lee et al. (2014) later used this method in collaborative filtering to estimate the user-item rating matrix with a weighted combination of local matrices. To our knowledge, our work is the first to propose local low-rank models for CL.

## 3 PROBLEM FORMULATION

In this section, we first formally state the problem formulation of continual learning, and then introduce the basic orthogonal-projection-based approach for continual learning and its fundamental computational complexity challenge.

**1) Continual Learning:** Continual learning (CL) considers a set of tasks $\mathbb{T} = \{t\}_{t=0}^{T}$ that arrive sequentially. Each task $t$ is associated with a dataset $\mathcal{D}_t = \{(\mathbf{x}_{t,i}, \mathbf{y}_{t,i})\}_{i=1}^{N^t}$ that contains $N^t$ samples, where $\mathbf{x}_{t,i}$ and $\mathbf{y}_{t,i}$ are the $i$-th datapoint and its label in task $t$. In this paper, we consider a fixed capacity neural network with $L$ layers, with weights being denoted as $\{\mathbf{W}^l\}_{l=1}^{L}$, where $\mathbf{W}^l$ is the layer-wise weight for the $l$-th layer. We let $\mathbf{x}_{t,i}^l$ denote the input of layer $l$, with $\mathbf{x}_{t,i}^1 = \mathbf{x}_{t,i}$. The output of layer $l$, which is also the input of layer $l+1$, is computed as $\mathbf{x}_{t,i}^{l+1} = f(\mathbf{W}^l, \mathbf{x}_{t,i}^l)$, where $f(\cdot)$ denotes the processing at layer $l$. In this paper, we focus on the CL setting, where we only have the access to the dataset of the new task $\mathcal{D}_t$ and no data samples of old tasks $j \in [0, t-1]$ are available. We denote the loss function as $\mathcal{L}(\mathbf{W}, \mathcal{D}_t) = \mathcal{L}_t(\mathbf{W})$, where $\mathbf{W}$ denotes the weights for the neural network model. To learn task $t$ in CL, we have the weights $\mathbf{W}_{t-1}$ after learning for task $t-1$. The purpose of CL is to learn the new task $t$ based on the weights in $\mathbf{W}_{t-1}$ and the new data $\mathcal{D}_t$.

**2) Orthogonal-Projection-Based Approach for CL:** To address the forgetting challenge in CL, there has been a recent line of works that propose model updating for the new task in the direction orthogonal to the subspace spanned by the old tasks' input. As an illustration of this basic idea, let the subspace spanned by the inputs of tasks 1's layer $l$ be denoted as $D_1^l$. The learnt model for task 1 is denoted as $\{\mathbf{W}_1^l\}_l^L$. To learn task 2, the current model $\mathbf{W}_1^l$ will be updated in a direction orthogonal to $D_1^l$. Let $\Delta \mathbf{W}_1^l$ denote the model update after learning task 2. It follows from the orthogonal direction that $\Delta \mathbf{W}_1^l \mathbf{x}_{1,i}^l = 0$. Also, after learning task 2, the model is $\mathbf{W}_2^l = \mathbf{W}_1^l + \Delta \mathbf{W}_1^l$. Thus, we have $\mathbf{W}_2^l \mathbf{x}_{1,i}^l = (\mathbf{W}_1^l + \Delta \mathbf{W}_1^l)\mathbf{x}_{1,i}^l = \mathbf{W}_1^l \mathbf{x}_{1,i}^l + \Delta \mathbf{W}_1^l \mathbf{x}_{1,i}^l = \mathbf{W}_1^l \mathbf{x}_{1,i}^l$, which implies that there is *no* interference to task 1 after the learning of task 2, hence avoiding "forgetting."

**3) Orthogonal-Projection-Based CL Approach with Backward Knowledge Transfer:** Although orthogonal-projection-based approaches can effectively address the forgetting problem, forward and backward knowledge transfers are impossible due to the restriction of model updates only in the subspace orthogonal to the input space of old tasks. To address this limitation, a trust region approach is proposed in (Lin et al., 2022b), which is built upon the following definitions(Lin et al., 2022b):

**Definition 1** (Sufficient Projection (Lin et al., 2022b)). For any new task $t \in [1, T]$, we say it has sufficient gradient projection on the input subspace of of old task $j \in [0, t-1]$ if for some $\lambda_1 \in (0, 1)$: $\|\mathrm{Proj}_{S_j}(\nabla \mathcal{L}_t(\mathbf{W}_{t-1}))\|_2 \geq \lambda_1' \|\nabla \mathcal{L}_t(\mathbf{W}_{t-1})\|_2$.

Here, $\mathrm{Proj}_{S_j}(\mathbf{A}) = \mathbf{S}_j(\mathbf{S}_j)^\top \mathbf{A}$ denotes the projection onto the input subspace $D_j$ of task $j$, where $\mathbf{S}_j$ is the basis of $D_j$. The definition of sufficient projection implies that tasks $t$ and $j$ have sufficient common bases between their input subspaces and hence strong correlation. While sufficient condition suggests strong correlation between tasks $t$ and $j$, a stronger condition suggesting positive correlation between tasks is also introduced in (Lin et al., 2022a) as follows:

**Definition 2** (Positive Correlation (Lin et al., 2022b)). A new task $t \in [1, T]$ has a positive correlation with an old task $j \in [0, t-1]$ if for some $\lambda_2 \in (0, 1)$, it holds that $\langle \nabla \mathcal{L}_j(\mathbf{W}_j), \nabla \mathcal{L}_t(\mathbf{W}_{t-1}) \rangle \geq \lambda_2 \|\nabla \mathcal{L}_j(\mathbf{W}_j)\|_2 \|\nabla \mathcal{L}_t(\mathbf{W}_{t-1})\|_2$.

Based on Definitions 1 and 2, the model space can be partitioned into three regimes (Lin et al., 2022a), where three different layer-wise update rules are applied:

- *Regime 1 (Forget Mitigation)* $\|\mathrm{Proj}_{D_j^l}(\nabla\mathcal{L}_t(\mathbf{W}_{t-1}^l))\|_2 \leq \lambda_1\|\nabla\mathcal{L}_t(\mathbf{W}_{t-1}^l)\|_2$: Due to the weak correlation between tasks in this regime, the model is updated based on orthogonal projection:

$$\nabla\mathcal{L}_t(\mathbf{W}^l) \leftarrow \nabla\mathcal{L}_t(\mathbf{W}^l) - \mathrm{Proj}_{D_j^l}(\nabla\mathcal{L}(\mathbf{W}^l)). \tag{1}$$

- *Regime 2 (Forward Knowledge Transfer):* A task $j$'s layer $l$ falls into Regime 2 if sufficient projection holds while positive correlation is not satisfied. Due to the potential "negative correlation" in this regime, forgetting still needs to be avoided by using orthogonal projection. However, thanks to the correlation between tasks, one can facilitate forward knowledge transfer. Putting both ideas together, the update rule in Regime 2 can be written as:

$$\nabla\mathcal{L}_t(\mathbf{W}^l) \leftarrow \nabla\mathcal{L}_t(\mathbf{W}^l) - \mathrm{Proj}_{D_j^l}(\nabla\mathcal{L}(\mathbf{W}^l)), \tag{2}$$
$$\mathbf{Q}_{j,t}^l \leftarrow \mathbf{Q}_{j,t}^l - \beta\nabla_{\mathbf{Q}}\mathcal{L}_t(\mathbf{W}^l - \mathrm{Proj}_{D_j^l}(\mathbf{W}^l) + \mathbf{W}^l\mathbf{S}_j^l\mathbf{Q}_{j,t}^l(\mathbf{S}_j^l)^\top),$$

where $\mathbf{S}_j^l$ is the basis matrix for subspace $D_j^l$ and $\mathbf{Q}_{j,t}^l$ is a diagonal scaling matrix to facilitate forward knowledge transfer (see Lin et al. (2022a;b) for details).

- *Regime 3 (Backward Knowledge Transfer):* A task $j$'s layer $l$ falls into Regime 3 if both sufficient projection and positive correlation conditions are satisfied. Due to the positive correlation between tasks, one can use a simple gradient-descent-type rule (with $\lambda$-regularization) to perform model update, which also helps improve the performances of old tasks (i.e., backward knowledge transfer):

$$\mathbf{W}^l \leftarrow \mathbf{W}^l - \alpha\nabla[\mathcal{L}_t(\mathbf{W}^l) + \theta\|\mathrm{Proj}_{D_j^l}(\mathbf{W}^l - \mathbf{W}_{t-1}^l)\|].$$

**4) Limitations and Challenges of Orthogonal-Projection-Based Approaches:** Although the aforementioned orthogonal-projection-based approaches (with forward-backward knowledge transfer) could effectively avoid forgetting without needing data from any old tasks, a *major challenge* in such approaches stems from checking the sufficient projection condition, which typically requires performing singular value decomposition (SVD) operations. It is well-known that SVD has an $O(n^3)$ computational complexity, which increases rapidly as $d$ increases. Thus, as the model size increases (e.g., in large-scale transformer models), computing SVD is expensive or even intractable. This limitation motivates us to develop efficient methods with low computation complexity for orthogonal-projection-based approaches in the subsequent section.

## 4 THE LOCAL MODEL SPACE PROJECTION APPROACH

In this section, we first introduce the basic idea of local representation and task subspace construction in Section 4.1, based on which we define task similarity with local projection in Section 4.2. These key notions allow us to further propose update rules based on local representations and task subspaces in Section 4.3. Lastly, we conduct theoretical performance analysis for our proposed LMSP-based orthogonal projection approach in Section 4.4.

### 4.1 LOCAL REPRESENTATION AND TASK SPACE CONSTRUCTION

As mentioned in Section 1, the basic idea of our LSMP approach to lower the SVD computational costs in orthogonal-projection-based CL approaches is based on a "divide and orthogonalize" principle. Our LMSP approach is built upon the following key notion of local representation. Specifically, given $N^j$ samples in an old task $j \in [0, t-1]$, we construct a representation matrix $\mathbf{R}_j^l = [\mathbf{r}_{j,1}^l, ...\mathbf{r}_{j,N^j}^l] \in \mathbb{R}^{M \times N^j}$ for layer $l$, where $M$ is the representation dimension and each $\mathbf{r}_{j,i}^l \in \mathbb{R}^M, i = 1, 2, ..., N^j$ is the representation of layer $l$ by forwarding the sample datapoint $\mathbf{x}_{j,i}$ through the model. Instead of directly applying SVD to the representation matrix $\mathbf{R}_j^l$, we approximate the matrix by a set of low-rank matrices defined by a set of anchor points. Inspired by (Lee et al., 2013), we define a smoothing kernel $K_h(s_1, s_2)$ with bandwidth $h$, where $(s_1, s_2) \in [M] \times [N^j]$ is an entry in the representation matrix $\mathbf{R}_j^l$. We denote by $\mathbf{K}_h^{(a,b)}$ the matrix whose $(i, j)$-th entry is $K_h((a, b), (i, j))$.

To obtain a set of local representation matrices, we first sample $m$ "anchor points" from the global representation matrix $\mathbf{R}_j^l$, which are denoted as $\{s_q \triangleq (i_q, j_q)\}_{q=1}^m$, where $(i_q, j_q) \in [M] \times [N^j]$ is the entry location of the $q$-th anchor point. It follows from (Wand & Jones, 1994; Lee et al., 2013) that the global representation matrix $\mathbf{R}_j^l$ has a locally low-rank structure and thus could be approximated by these local representation matrices $\{\hat{\mathbf{R}}_j^l(s_q)\}_{q=1}^m$ corresponding to these anchor points (i.e., Nadaraye-Waston regression):

$$\mathbf{R}_j^l \approx \hat{\hat{\mathbf{R}}}_j^l \triangleq \sum_{q=1}^m \frac{K_h(s_q, s)}{\sum_{p=1}^m K_h(s_p, s)} \hat{\mathbf{R}}_j^l. \tag{3}$$

To obtain the local representation matrices $\{\hat{\mathbf{R}}_j^l(s_q)\}_{q=1}^m$ in Eq. (3), we adopt a product form for the general kernel function $K_h(s_1, s_2) = K_h((a, b), (c, d)) = K_{h_1}(a, c)K'_{h_2}(b, d)$, where $s_1, s_2 \in [M] \times [N^j]$ and $K, K'$ are kernels on the spaces $[M]$ and $[N^j]$, respectively. We summarize several popular smoothing kernels in Appendix C. In this paper, we use the Gaussian kernel for both $K, K'$ (we will conduct ablation studies in Section 5). There are two ways to choose the anchor points $\{s_q \triangleq (i_q, j_q)\}_{q=1}^m$: 1) sample uniformly at random from the representation matrix in $[M] \times [N^j]$; 2) use $K$-means or other clustering methods to pre-cluster the representation matrix and then use their centers as the anchor points. In our numerical studies, we do not observe a significant difference between these two methods. For simplicity, we use the random sample strategy in our experiments.

Next, with local representations, next, we will show how the local model spaces are constructed for task $j$ at layer $l$. For an old task $j \in [0, t-1]$, to obtain the basis $\mathbf{S}_j^l$ at layer $l$, traditional methods (Saha et al., 2021; Lin et al., 2022b) adopted the standard singular value decomposition (SVD) for the representation matrix of each layer, which incurs a high computation cost of $\mathcal{O}(MN^j \min(M, N^j)) = \mathcal{O}(n^3)$. In contrast, since each local model has a low-rank structure, the computation can be significantly reduced. Specifically, we first obtain the local decomposed matrices $\mathbf{A}, \mathbf{B}$ for each anchor point $s_q$ by minimizing the following global least square loss in Eq. (4):

$$\{(\mathbf{A}^{(q)}, \mathbf{B}^{(q)})\}_{q=1}^m :=$$
$$\underset{\mathbf{A}^{(q)}, \mathbf{B}^{(q)}}{\arg\min} \sum_{x,y \in \Omega} \left[ \sum_{q=1}^m \left( \frac{K_h^{(q)} \odot [\mathbf{A}^{(q)}\mathbf{B}^{(q)\top}]}{\sum_{p=1}^m K_h^{(p)}} - \mathbf{R}_j^l \right)^2 \right]_{x,y} + \sum_{q=1}^m [\lambda_A^{(q)} \|\mathbf{A}^{(q)}\|_F^2 + \lambda_B^{(q)} \|\mathbf{B}^{(q)}\|_F^2], \tag{4}$$

where $K_h^{(q)} = K_h^{s_q} = K_h^{(i_q, j_q)}$ is the kernel matrix whose $(a, b)$-th entry is $K_h((i_q, j_q), (a, b)) = K_{h_1}(i_q, a)K'_{h_2}(j_q, b)$ and $\odot$ is the Hadamard product. We also add $\ell_2$ regularization as is standard in conventional SVD. Similar to (Lee et al., 2013), we can execute the algorithm in a parallel fashion:

$$(\mathbf{A}^{(q)}, \mathbf{B}^{(q)}) := \underset{\mathbf{A}, \mathbf{B}}{\arg\min} \sum_{x,y \in \Omega} [K_h^{(q)} \odot ([\mathbf{A}\mathbf{B}^\top] - \mathbf{R}_j^l)^2]_{x,y} + \lambda_A \|\mathbf{A}\|_F^2 + \lambda_B \|\mathbf{B}\|_F^2.$$

As a variant of low-rank matrix completion, this problem can be solved efficiently via various methods, including AltMin (Jain et al., 2013; Hastie et al., 2015), singular value projection (Netrapalli et al., 2014; Jain et al., 2010), Riemannian GD (Wei et al., 2016), ScaledGD (Tong et al., 2021; Xu et al., 2023), etc; see (Chen & Chi, 2018; Chi et al., 2019) for recent overviews. In this paper, we use the AltMin method to find the optimizer and obtain the basis for each local model. Denote the rank for each local model as $r \ll \min(M, N^j)$, and $\mathbf{A} \in \mathbb{R}^{M \times r}, \mathbf{B} \in \mathbb{R}^{N^j \times r}$. Later we adopt QR decomposition for $\mathbf{A} = \hat{\mathbf{U}}\mathbf{\Omega}_A, \mathbf{B} = \hat{\mathbf{V}}\mathbf{\Omega}_B$, where $\mathbf{\Omega}_A, \mathbf{\Omega}_B \in \mathbb{R}^{r \times r}$, and then perform SVD on the $r \times r$ matrix to achieve: $\mathbf{\Omega}_A \mathbf{\Omega}_B^\top = \mathbf{U}_\Omega \mathbf{\Sigma} \mathbf{V}_\Omega^\top$. The final basis for local model space $q$ can be constructed as $\{\mathbf{S}_j^{l,(q)} \triangleq \hat{\mathbf{U}}_j^{l,(q)} \mathbf{U}_{\Omega,j}^{l,(q)}\}_{q=1}^m \in \mathbb{R}^{M \times r}$.

Then, for a new task $t$, we treat all $m$ local model spaces as $m$ old tasks. As a result, we have a total of $tm$ old tasks as candidates for new task $t$ to find the top-$k$ correlated ones. Since the AltMin algorithm has the complexity of $\mathcal{O}(MN^j r) = \mathcal{O}(n^2)$, the total complexity can be reduced to $\mathcal{O}(n^2 m) = \mathcal{O}(n^2)$, as the total number of anchor points $m \ll \min(M, N^j)$. Thus the computation cost is significantly reduced.

## 4.2 TASK SIMILARITY WITH LOCAL PROJECTION

With the local representations in Section 4.1, we are now in a position to introduce the following definitions on task gradients to formally characterize the task similarity. Toward this end, we need the following definitions that generalize Definitions 1 and 2 to local settings:

**Definition 3** (Local Sufficient Projection). For any new task $t \in [1, T]$, we say it has local sufficient gradient projection on the local subspace $q \in [1, m]$ of old task $j \in [0, t-1]$ if for some $\lambda_1 \in (0, 1)$: $\|\mathrm{Proj}_{K_h^{(q)} D_j}(\nabla \mathcal{L}_t(\mathbf{W}_{t-1}))\|_2 \geq \lambda_1 \|\nabla \mathcal{L}_t(\mathbf{W}_{t-1})\|_2$.

**Definition 4** (Local Positive Correlation). For any new task $t \in [1, T]$, we say that it has local positive correlation with the local subspace $q \in [1, m]$ of old task $j \in [0, t-1]$ if for some $\lambda_2 \in (0, 1)$: $\langle \nabla \mathcal{L}_j^{(q)}(\mathbf{W}_j^{(q)}), \nabla \mathcal{L}_t(\mathbf{W}_{t-1}) \rangle \geq \lambda_2 \|\nabla \mathcal{L}_j^{(q)}(\mathbf{W}_j^{(q)})\|_2 \|\nabla \mathcal{L}_t(\mathbf{W}_{t-1})\|_2$.

Here, for any matrix $\mathbf{A}$, $\mathrm{Proj}_{K_h^{(q)} D_j}(\mathbf{A}) \triangleq \mathbf{S}_j^{(q)} \mathbf{S}_j^{(q)\top} \mathbf{A}$ defines the projection on the input local model space for anchor point $q$ of old task $j$, and $\mathbf{S}^{(q)}_j$ is the basis for this local model space. Compared to Definition 1, the projection space in Definition 3 is changed to the $q$-th local model basis rather than the global basis for task $j$. Definition 3 implies that task $t$ and the $q$-th local model of task $j$ have sufficient common bases and are strongly correlated since the gradient lies in the span of the input Zhang et al. (2021). Also, similar to Definition 2, Definition 4 goes one step further to characterize the task similarity. In addition to local sufficient projection and positive correlation conditions, we propose a *new* notion called "local relative orthogonality" as follows:

**Definition 5** (Local Relative Orthogonality). For any new task $t \in [1, T]$, we say it is more locally relative orthogonal to local subspace $q \in [1, m]$ of old task $j \in [0, t-1]$ than the global subspace old task $j \in [0, t-1]$ for some $\lambda_3 \in (0, 1)$ if: $\|\mathrm{Proj}_{K_h^{(q)} D_j}(\nabla \mathcal{L}_t(\mathbf{W}_{t-1}))\|_2 = \lambda_3 \|\mathrm{Proj}_{D_j}(\nabla \mathcal{L}_t(\mathbf{W}_{t-1}))\|_2 \leq \|\mathrm{Proj}_{D_j}(\nabla \mathcal{L}_t(\mathbf{W}_{t-1}))\|_2$.

The local relative orthogonality means that the input of the $q$-th local model space for old task $j$ is more orthogonal to the new task $t$ than the global one, which indicates that updating the model along the $\nabla \mathcal{L}_t(\mathbf{W})$ direction would not introduce less inference to old task $j$, thus mitigating the forgetting problem. Note that Definitions 4–5 characterize the similarity based on the old model weights $\mathbf{W}_{t-1}$, hence they allow the task similarity detection before learning the new task $t$.

## 4.3 LOW-COMPLEXITY CONTINUAL LEARNING WITH LOCAL MODEL SPACE PROJECTION

With the local representations and the associated task similarity, we propose the following LMSP-based orthogonal projection approach in the spirit of CUBER in Section 3 aiming to avoid forgetting while enabling backward knowledge transfer. Specifically, based on Definitions 3 and 4, we have these following regimes:

**Regime 1** (Forget Mitigation): For a new task $t$'s layer $l$, if $\|\mathrm{Proj}_{K_h^{(q)} D_j^l}(\nabla \mathcal{L}_t(\mathbf{W}_{t-1}^l))\|_2 < \lambda_1 \|\nabla \mathcal{L}_t(\mathbf{W}_{t-1}^l)\|_2$, we say the $q$-th local model of old task $j$ falls in Regime 1. Note that in this case, since task $t$ and task $j^{(q)}$ are *relatively orthogonal,* we update the model in the direction of orthogonal projection to avoid forgetting:

$$\nabla \mathcal{L}_t(\mathbf{W}^l) \leftarrow \nabla \mathcal{L}_t(\mathbf{W}^l) - \mathrm{Proj}_{K_h^{(q)} D_j^l}(\nabla \mathcal{L}_t(\mathbf{W}^l)). \tag{5}$$

**Regime 2** (Forward Knowledge Transfer): For a new task $t$'s layer $l$, if it holds that $\|\mathrm{Proj}_{K_h^{(q)} D_j^l}(\nabla \mathcal{L}_t(\mathbf{W}_{t-1}^l))\|_2 \geq \lambda_1 \|\nabla \mathcal{L}_t(\mathbf{W}_{t-1}^l)\|_2$ and $\langle \nabla \mathcal{L}_j^{(q)}(\mathbf{W}_j^{l,(q)}), \nabla \mathcal{L}_t(\mathbf{W}_{t-1}^l) \rangle < \lambda_2 \|\nabla \mathcal{L}_j^{(q)}(\mathbf{W}_j^{l,(q)})\|_2 \|\nabla \mathcal{L}_t(\mathbf{W}_{t-1}^l)\|_2$, we say the $q$-th local model of old task $j$ falls into Regime 2. In this case, since task $t$ and task $j^{(q)}$ are strongly correlated on gradient norm projection but negatively correlated on gradient direction, we still update the model on the orthogonal projection and use a scalar matrix $\mathbf{Q}$ to facilitate forward knowledge similar to the idea in (Lin et al., 2022b):

$$\nabla \mathcal{L}_t(\mathbf{W}^l) \leftarrow \nabla \mathcal{L}_t(\mathbf{W}^l) - \mathrm{Proj}_{K_h^{(q)} D_j^l}(\nabla \mathcal{L}_t(\mathbf{W}^l)), \tag{6}$$

$$\mathbf{Q}_{j,t}^{l,(q)} \leftarrow \mathbf{Q}_{j,t}^{l,(q)} - \beta \nabla_{\mathbf{Q}} \mathcal{L}_t(\mathbf{W}^l - \mathrm{Proj}_{K_h^{(q)} D_j^l}(\mathbf{W}^l) - \mathbf{W}^l \mathbf{S}_j^{l,(q)} \mathbf{Q}_{j,t}^{l,(q)} \mathbf{S}_j^{l,(q)\top}).$$

**Regime 3** (Backward Knowledge Transfer): For a new task $t$'s layer $l$, if it holds that $\|\mathrm{Proj}_{K_h^{(q)} D_j^l}(\nabla \mathcal{L}_t(\mathbf{W}_{t-1}^l))\|_2 \geq \lambda_1 \|\nabla \mathcal{L}_t(\mathbf{W}_{t-1}^l)\|_2$ and $\langle \nabla \mathcal{L}_j^{(q)}(\mathbf{W}_j^{l,(q)}), \nabla \mathcal{L}_t(\mathbf{W}_{t-1}^l) \rangle \geq \lambda_2 \|\nabla \mathcal{L}_j^{(q)}(\mathbf{W}_j^{l,(q)})\|_2 \|\nabla \mathcal{L}_t(\mathbf{W}_{t-1}^l)\|_2$, we say the $q$-th local model of old task $j$ falls into Regime 3.

---

**Algorithm 1** Efficient Continual Learning with Local Model Space Projection (LMSP)

---

1: Input: task sequence $\mathbb{T} = \{t\}_{t=0}^T$;
2: Learn first $j \in [0, t-1]$ task using vanilla stochastic gradient descent;
3: **for** each old task $j$ **do**
4:     Sample $m$ anchor point
5:     Extract basis $\mathbf{S}_j^{l,(q)}$ for each local model space $q$ using the learnt model $\mathbf{W}_j$
6: **end for**
7: **for** each new task $t$ **do**
8:     Calculate gradient $\nabla \mathcal{L}_t(\mathbf{W}_{t-1})$;
9:     Evaluate the *local sufficient projection* and *local positive correlation* conditions for layer-wise correlation computation to determine its membership in $\mathrm{Reg}_{t,1}^l$, $\mathrm{Reg}_{t,2}^l$ or $\mathrm{Reg}_{t,3}^l$;
10:     **for** $k = 1, 2, ...$ **do**
11:         Update the model and scaling matrices by solving Eq. (7);
12:     **end for**
13: **end for**
14: Output: The learnt model $\mathbf{W}_t$, scaling matrices $\{\mathbf{Q}_{j,t}^{l,(q)}\}_{l,j^{(q)} \in \mathrm{Reg}_{t,3}^l \bigcup \mathrm{Reg}_{t,3}^l}$;

---

In this case, since task $t$ and task $j^{(q)}$ are positively correlated in both norm and direction, updating the model directly along with $\nabla \mathcal{L}_t(\mathbf{W}^l)$ could not only lead to a better model for continual learning, but also improve the performance of old task $j$. Since the weight projection is frozen, i.e., $\mathrm{Proj}_{K_h^{(q)} D_j^l}(\mathbf{W}_{t-1}^l) = \mathrm{Proj}_{K_h^{(q)} D_j^l}(\mathbf{W}_j^l)$, we update the model as follows:

$$\mathbf{W}^l \leftarrow \mathbf{W}^l - \alpha \nabla [\mathcal{L}_t(\mathbf{W}^l) + \theta \|\mathrm{Proj}_{K_h^{(q)} D_j^l}(\mathbf{W}^l - \mathbf{W}_{t-1}^l)\|].$$

In summary, the optimization problem for learning a new task $t$ can be written as follows:

$$\min_{\mathbf{W}, \{\mathbf{Q}_{j,t}^{l,(q)}\}_{l,j^{(q)} \in \mathrm{Reg}_{t,3}^l \bigcup \mathrm{Reg}_{t,3}^l}} \mathcal{L}_t(\{\tilde{\mathbf{W}}^l\}_l) + \theta \sum_l \sum_{j^{(q)} \in \mathrm{Reg}_{t,3}^l} \|\mathrm{Proj}_{K_h^{(q)} D_j^l}(\mathbf{W}^l - \mathbf{W}_{t-1}^l)\|, \tag{7}$$

$$s.t. \ \tilde{\mathbf{W}}^l = \mathbf{W}^l + \sum_{j^{(q)} \in \mathrm{Reg}_{t,2}^l \bigcup j^{(q)} \in \mathrm{Reg}_{t,3}^l} [\mathbf{W}^l \mathbf{S}_j^{l,(q)} \mathbf{Q}_{j,t}^{l,(q)} \mathbf{S}_j^{l,(q)\top} - \mathrm{Proj}_{K_h^{(q)} D_j^l}(\mathbf{W}^l)],$$

$$\nabla \mathcal{L}_t(\mathbf{W}^l) = \nabla \mathcal{L}_t(\mathbf{W}^l) - \sum_{j^{(q)} \in \mathrm{Reg}_{t,1}^l \bigcup j^{(q)} \in \mathrm{Reg}_{t,2}^l} \mathrm{Proj}_{K_h^{(q)} D_j^l}(\nabla \mathcal{L}_t(\mathbf{W}^l)).$$

Since task similarity is calculated before learning the new task $t$, we first determine the regimes for different local model space from old task $j$, and then directly update the model for tasks in Regime 3 while using orthogonal projection to preserve the knowledge for the rest. The scaled weight projection is used for old tasks in both Regime 2 and Regime 3 to facilitate forward knowledge transfer. The overview of our LMSP-based efficient continual learning framework is described in Algorithm 1.

## 4.4 THEORETICAL PERFORMANCE ANALYSIS

In this subsection, we will establish the convergence rate and backward knowledge transfer of our proposed LMSP-based orthogonal projection approach. Without loss of generality, consider the scenario of learning two consecutive tasks 1 and 2. Note that since (Lin et al., 2022a) has already conducted theoretical analysis for the vanilla GD-type update (cf. Rule #2 in Lin et al. (2022a)), which is also applicable in our work, we will only focus on the major difference in our work, which lies in the analysis for the local and global orthogonal-projection-based updates. For simplicity, considering the scenario with a sequence of two tasks 1 and 2. Let $\mathcal{F}(\mathbf{W}) = \mathcal{L}(\mathbf{W}, \mathcal{D}_1) + \mathcal{L}(\mathbf{W}, \mathcal{D}_2)$, $\boldsymbol{g}_1(\mathbf{W}) = \nabla_{\mathbf{W}} \mathcal{L}(\mathbf{W}, \mathcal{D}_1)$ and $\boldsymbol{g}_2(\mathbf{W}) = \nabla_{\mathbf{W}} \mathcal{L}(\mathbf{W}, \mathcal{D}_2)$. Given $\bar{\boldsymbol{g}}(\mathbf{W}^{(k)}) = \boldsymbol{g}(\mathbf{W}^{(k)}) - \mathrm{Proj}_{K_h^{(q)} D_j}(\boldsymbol{g}(\mathbf{W}^{(k)}))$ as the gradients for the local orthogonal-projection-based updates in Eq. (5) as well as Eq. (6), and $\ddot{\boldsymbol{g}}(\mathbf{W}^{(k)}) = \boldsymbol{g}(\mathbf{W}^{(k)}) - \mathrm{Proj}_{D_j}(\boldsymbol{g}(\mathbf{W}^{(k)}))$ as the gradients for the global orthogonal-projection-based updates in Eq. (1) as well as Eq. (2), we denote step $k \in [0, K-1]$ and the model learned parameters for task 1 $\mathbf{W}_1 = \mathbf{W}^{(0)}$ as the initialization of the new task model weights. We first state our major convergence rate result for orthogonal-projection-based update as follows:

**Theorem 1.** *Suppose loss $\mathcal{L}$ is $B$-Lipschitz and $\frac{H}{2}$-smooth. Let $\alpha \leq \min\{\frac{1}{H}, \frac{\gamma\|\bar{\boldsymbol{g}}_1(\mathbf{W}^{(0)})\|}{HBK}\}$ and $\lambda_1 \geq \sqrt{1 - 2\frac{2\|\bar{\boldsymbol{g}}_2(\mathbf{W}^{(0)})\| - \|\bar{\boldsymbol{g}}_1(\mathbf{W}^{(0)})\|}{\gamma^2\|\bar{\boldsymbol{g}}_1(\mathbf{W}^{(0)})\|}}$ for some $\gamma \in (0,1)$. We have the following results:*

*(1) if $\mathcal{L}$ is convex, the orthogonal-projection-based update in Regimes 1 and 2 for task 2 converges to the optimal model $\mathbf{W}^{\star} = \arg\min \mathcal{F}(\mathbf{W})$;*

*(2) if $\mathcal{L}$ is non-convex, the orthogonal-projection-based update in Regimes 1 and 2 for task 2 converges to the first order stationary point, i.e., $\min_k \|\nabla\mathcal{F}(\mathbf{W}^{(k)})\|^2 \leq \frac{2}{\alpha K}\sum_{k=0}^{K-1}[\mathcal{F}(\mathbf{W}^{(k)}) - \mathcal{F}(\mathbf{W}^{\star})] + \frac{[2+\gamma^2(5-\lambda_1^2)]}{2}\|\bar{\boldsymbol{g}}_1(\mathbf{W}^{(0)})\|^2 + 4\|\boldsymbol{g}_1(\mathbf{W}^{(0)})\|^2 + 4\|\boldsymbol{g}_2(\mathbf{W}^{(0)})\|^2.$*

Theorem 1 characterizes the convergence of the joint objective function $\mathcal{F}(\mathbf{W})$ when updating the model with orthogonal-projection-based updates in the convex setting, as well as the convergence to a first-order stationary point in the non-convex setting when the $q$-th local model of task 1 and task 2 satisfy the local sufficient projection definition with certain $\lambda_1$. Hence, it finally benefits the joint learning of task 1 and 2. The proof of Theorem 1 is relegated to Appendix A due to space limitation. The next result establishes the backward knowledge transfer of our proposed CL approach:

**Theorem 2.** *Suppose loss $\mathcal{L}$ is $B$-Lipschitz and $\frac{H}{2}$-smooth. We have the following results:*

*(1) Let $\mathbf{W}^s$ and $\mathbf{W}^c$ be the model parameters after one update to an initial model $\mathbf{W}$ by using local and global orthogonal-projection-based updates, respectively. Suppose the new task satisfy local relative orthogonality for a $\lambda_3 \in (0,1)$, i.e., $\|Proj_{K_h^{(q)}D_1}(\boldsymbol{g}_2(\mathbf{W}^{(i)}))\|_2 = \lambda_3\|Proj_{D_1}(\boldsymbol{g}_2(\mathbf{W}^{(i)}))\|_2$ for $i \in [0, k-1]$, $\alpha \leq \min\{\frac{1}{H}, \frac{\gamma\|\bar{\boldsymbol{g}}_1(\mathbf{W}^{(0)})\|}{HBK}\}$ and $\lambda_1 \geq \max\{\sqrt{1 - 2\frac{2\|\bar{\boldsymbol{g}}_2(\mathbf{W}^{(0)})\| - \|\bar{\boldsymbol{g}}_1(\mathbf{W}^{(0)})\|}{\gamma^2\|\bar{\boldsymbol{g}}_1(\mathbf{W}^{(0)})\|}}, \sqrt{1 - \frac{(1-\lambda_3^2)(2+\alpha H)\lambda_1'^2}{1+2\alpha H}}\}$, then we have $\mathcal{F}(\mathbf{W}^s) \leq \mathcal{F}(\mathbf{W}^c)$;*

*(2) Let $\mathbf{W}^{(k)}$ be the $k$-th iterate for task 2 with the $\theta$-regularized update in Regime 3. Suppose that $\alpha \leq \frac{4\|\bar{\boldsymbol{g}}_1(\mathbf{W}^{(0)})\|}{HBk^{1.5}}$. It follows that $\mathcal{L}_1(\mathbf{W}^{(k)}) \leq \mathcal{L}_1(\mathbf{W}_1) = \mathcal{L}_1(\mathbf{W}^{(0)}).$*

The first claim in Theorem 2 indicates that updating the model using the local orthogonal-projection-based updates achieves lower loss value than the global orthogonal-projection-based updates when the $q$-th local model of task 1 and task 2 satisfy the sufficient projection with some $\lambda_1$ and the local relative orthogonality in Definition 5 with some $\lambda_3$. The second claim in Theorem 2 suggests that the local orthogonal-projection-based update results in a better model for task 1 with respect to $\mathcal{L}_1$. The proofs of Theorem 2 is also relegated to Appendix B due to space limitation.

## 5 NUMERICAL RESULTS

**1) Datasets:** We evaluate the performance of our LMSP on four public datasets for CL: (1) Permuted MNIST (LeCun et al., 2010); (2) CIFAR-100 Split (Krizhevsky et al., 2009); (3) 5-Datasets (Lin et al., 2022a;b); and (4) MiniImageNet (Vinyals et al., 2016). Due to space limitation, the detailed information of these datasets is relegate to Appendix D.

**2) Baseline Methods:** We compare our LMSP method with the following baseline methods: (1) *EWC* (Kirkpatrick et al., 2017b): EWP adopts the Fisher information matrix for weights importance evaluation. (2) *HAT* (Serra et al., 2018): HAT preserves the knowledge of an old task by learning a hard attention mask; (3) *Orthogonal Weight Modulation (OWM)* (Zeng et al., 2019): OWM projects the gradient of a new task to the orthogonal direction of the input subspace of an old task by learning a projector matrix; (4) *Gradient Projection Memory (GPM)* Saha et al. (2021): GPM first stores the basis of the input subspace of old tasks, and then use the gradient projection orthogonal to the subspace spanned by these stored bases to update the model; (5) *TRGP* (Lin et al., 2022b): TRGP uses a scaled weight projection to facilitate the forward knowledge transfer from related old tasks to the new task; (6) *CUBER* Lin et al. (2022a): CUBER categorizes the task correlation as strong projection and positive correlation. (7) *Averaged GEM (A-GEM)* (Chaudhry et al., 2018): A-GEM stores and incorporate old tasks' data in computing gradients for the new task's learning; (8) *Experience Replay with Reservoir sample (ER-Res)* Chaudhry et al. (2019a): ER-Res uses a small episodic memory to store old task samples to address the forgetting problem; and (9) *Multitask* (Saha et al., 2021): Multitask jointly learns all tasks once with a single network using all datasets.

Table 1: The ACC and BWT performance comparisons between LMSP (ours) and baselines.

| Method | PMNIST | | CIFAR-100 Split | | 5-Dataset | | MiniImageNet | |
|---|---|---|---|---|---|---|---|---|
| | ACC(%) | BWT(%) | ACC(%) | BWT(%) | ACC(%) | BWT(%) | ACC(%) | BWT(%) |
| Multitask | 96.70 | - | 79.58 | - | 91.54 | - | 69.46 | - |
| OWM | 90.71 | -1 | 50.94 | -30 | - | - | - | - |
| EWC | 89.97 | -4 | 68.80 | -2 | 88.64 | -4 | 52.01 | -12 |
| HAT | - | - | 72.06 | 0 | 91.32 | -1 | 59.78 | -3 |
| A-GEM | 83.56 | -14 | 63.98 | -15 | 84.04 | -12 | 57.24 | -12 |
| ER-Res | 87.24 | -11 | 71.73 | -6 | 88.31 | -4 | 58.94 | -7 |
| GPM | 93.91 | -3 | 72.48 | -0.9 | 91.22 | -1 | 60.41 | -0.7 |
| TRPG | 96.26 | -1.01 | 74.98 | -0.15 | 92.41 | -0.08 | **64.46** | -0.89 |
| CUBER | 97.04 | -0.11 | **75.29** | 0.14 | 92.85 | -0.13 | 63.67 | 0.11 |
| **LMSP** | **97.48** | **0.16** | 74.21 | **0.94** | **93.78** | **0.07** | 64.2 | **1.55** |

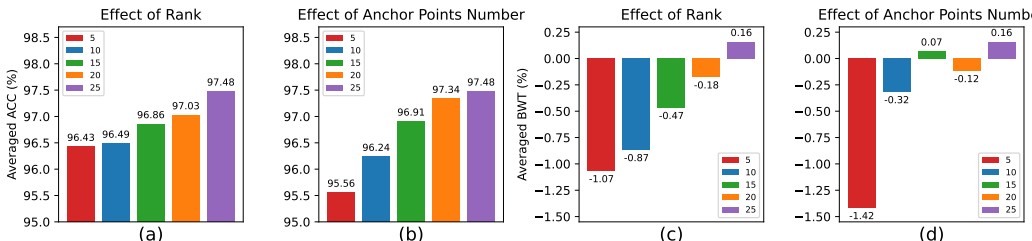

Figure 1: Ablation studies on rank and number of anchor points.

**3) Evaluation Metrics:** We use the following two metrics to evaluate the learning performance of the baseline models and our model: (1) Accuracy (ACC), which is the final averaged accuracy over all tasks; (2) Backward transfer (BWT), which is the averaged accuracy change of each task after learning the new task. $ACC = \frac{1}{T} \sum_{i=1}^{T} A_{T,i}$ and $BWT = \frac{1}{T-1} \sum_{i=1}^{T-1} (A_{T,i} - A_{i,i})$, where $A_{i,j}$ is the testing accuracy of task $j$ after learning task $i$.

**4) Experimental Results:** We can see from Table 1 that our LMSP method outperforms other baseline methods in both ACC and BWT. It is worth noting that the BWT performance in our method is generally better than CUBER. To understand the efficacy of the proposed techniques, we further conduct ablation studies. We show the effects with different rank values and number of anchor points for our approach in Fig. 1. Due to space limitation, we relegate the ablation study results with different kernel types to the appendix E.

*4-1) Effect of Low Rank*: Fig. 1(a)(c) shows the results of our method using different low rank value $r$. We can see that, as expected, the model's performance becomes better when the rank becomes higher. In general, a higher rank value implies less information loss during the bases construction. Further, as the rank value becomes sufficiently high, the performance improvement becomes insignificant since most of the information has already been included.

*4-2) Effect of Anchor Point Number*: Fig. 1(b)(d) illustrates the performance of our LMSP method with a different number of anchor points. We can see that more anchor points leads to better performance since more candidate old tasks are generated, thus it would be easier to find more correlated old tasks with the new task. However, as the number of anchor points increases, the computation cost also increases correspondingly, which implies a trade-off between performance and cost.

## 6 CONCLUSION

In this paper, we proposed a new efficient orthogonal-projection-based continual learning strategy based on local model space projection (LMSP), which not only reduces the complexity of basis computation, but also facilitates forward and backward knowledge transfers. We conducted theoretical analysis to show that the new task's performance could benefit from the local old tasks more than just using the global old task under certain circumstances. Our extensive experiments on public datasets demonstrated the efficacy of our approach. Future work includes deploying our efficient CL method to some popular deep learning structures such as transformers and LLMs, and extending our approach to more general CL settings.

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

# A  PROOF OF THEOREM 1

*Proof.* For a $\frac{H}{2}$-smooth loss function $\mathcal{L}$, it can be easily shown that $\mathcal{F}$ is $H$-smooth. (1) For any $k \in [0, K]$, we can have:

$$\mathcal{F}(\mathbf{W}^{(k+1)}) \leq \mathcal{F}(\mathbf{W}^{(k)}) + \nabla\mathcal{F}(\mathbf{W}^{(k)})^{\top}(\mathbf{W}^{(k+1)} - \mathbf{W}^{(k)}) + \frac{H}{2}\|\mathbf{W}^{(k+1)} - \mathbf{W}^{(k)}\|^2$$

$$= \mathcal{F}(\mathbf{W}^{(k)}) + (\boldsymbol{g}_1(\mathbf{W}^{(k)}) + \boldsymbol{g}_2(\mathbf{W}^{(k)}))^{\top}(-\alpha\bar{\boldsymbol{g}}_2(\mathbf{W}^{(k)})) + \frac{\alpha^2 H}{2}\|\bar{\boldsymbol{g}}_2(\mathbf{W}^{(k)})\|^2$$

$$= \mathcal{F}(\mathbf{W}^{(k)}) - [\alpha - \frac{\alpha^2 H}{2}]\|\bar{\boldsymbol{g}}_2(\mathbf{W}^{(k)})\|^2 - \alpha\langle\bar{\boldsymbol{g}}_1(\mathbf{W}^{(k)}), \bar{\boldsymbol{g}}_2(\mathbf{W}^{(k)})\rangle, \tag{8}$$

since:

$$\langle\boldsymbol{g}_1(\mathbf{W}^{(k)}), \bar{\boldsymbol{g}}_2(\mathbf{W}^{(k)})\rangle = \langle\text{Proj}_{K_h^{(q)}D_1}(\boldsymbol{g}_1(\mathbf{W}^{(k)})), \bar{\boldsymbol{g}}_2(\mathbf{W}^{(k)})\rangle + \langle\bar{\boldsymbol{g}}_1(\mathbf{W}^{(k)}), \bar{\boldsymbol{g}}_2(\mathbf{W}^{(k)})\rangle, \tag{9}$$

$$\langle\boldsymbol{g}_2(\mathbf{W}^{(k)}), \bar{\boldsymbol{g}}_2(\mathbf{W}^{(k)})\rangle = \langle\text{Proj}_{K_h^{(q)}D_1}(\boldsymbol{g}_2(\mathbf{W}^{(k)})), \bar{\boldsymbol{g}}_2(\mathbf{W}^{(k)})\rangle + \langle\bar{\boldsymbol{g}}_2(\mathbf{W}^{(k)}), \bar{\boldsymbol{g}}_2(\mathbf{W}^{(k)})\rangle, \tag{10}$$

and:

$$\langle\text{Proj}_{K_h^{(q)}D_1}(\boldsymbol{g}_1(\mathbf{W}^{(k)})), \bar{\boldsymbol{g}}_2(\mathbf{W}^{(k)})\rangle = 0, \tag{11}$$

$$\langle\text{Proj}_{K_h^{(q)}D_1}(\boldsymbol{g}_2(\mathbf{W}^{(k)})), \bar{\boldsymbol{g}}_2(\mathbf{W}^{(k)})\rangle = 0. \tag{12}$$

For the term $\langle\bar{\boldsymbol{g}}_2(\mathbf{W}^{(k)}), \bar{\boldsymbol{g}}_2(\mathbf{W}^{(k)})\rangle$, it follows that:

$$\langle\bar{\boldsymbol{g}}_1(\mathbf{W}^{(k)}), \bar{\boldsymbol{g}}_2(\mathbf{W}^{(k)})\rangle$$
$$= \langle\bar{\boldsymbol{g}}_1(\mathbf{W}^{(k)}) - \bar{\boldsymbol{g}}_1(\mathbf{W}^{(0)}) + \bar{\boldsymbol{g}}_1(\mathbf{W}^{(0)}), \bar{\boldsymbol{g}}_2(\mathbf{W}^{(k)})\rangle$$
$$= \langle\bar{\boldsymbol{g}}_1(\mathbf{W}^{(k)}) - \bar{\boldsymbol{g}}_1(\mathbf{W}^{(0)}), \bar{\boldsymbol{g}}_2(\mathbf{W}^{(k)})\rangle + \langle\bar{\boldsymbol{g}}_1(\mathbf{W}^{(0)}), \bar{\boldsymbol{g}}_2(\mathbf{W}^{(k)})\rangle$$
$$= \langle\bar{\boldsymbol{g}}_1(\mathbf{W}^{(k)}) - \bar{\boldsymbol{g}}_1(\mathbf{W}^{(0)}), \bar{\boldsymbol{g}}_2(\mathbf{W}^{(k)})\rangle + \langle\bar{\boldsymbol{g}}_1(\mathbf{W}^{(0)}), \bar{\boldsymbol{g}}_2(\mathbf{W}^{(k)}) - \bar{\boldsymbol{g}}_2(\mathbf{W}^{(0)})\rangle + \langle\bar{\boldsymbol{g}}_1(\mathbf{W}^{(0)}), \bar{\boldsymbol{g}}_2(\mathbf{W}^{(0)})\rangle. \tag{13}$$

Considering

$$2\langle\bar{\boldsymbol{g}}_1(\mathbf{W}^{(k)}) - \bar{\boldsymbol{g}}_1(\mathbf{W}^{(0)}), \bar{\boldsymbol{g}}_2(\mathbf{W}^{(k)})\rangle + \|\bar{\boldsymbol{g}}_1(\mathbf{W}^{(k)}) - \bar{\boldsymbol{g}}_1(\mathbf{W}^{(0)})\|^2 + \|\bar{\boldsymbol{g}}_2(\mathbf{W}^{(k)})\|^2$$
$$= \|\bar{\boldsymbol{g}}_1(\mathbf{W}^{(k)}) - \bar{\boldsymbol{g}}_1(\mathbf{W}^{(0)}) + \bar{\boldsymbol{g}}_2(\mathbf{W}^{(k)})\|^2 \geq 0, \tag{14}$$

we have:

$$\langle\bar{\boldsymbol{g}}_1(\mathbf{W}^{(k)}) - \bar{\boldsymbol{g}}_1(\mathbf{W}^{(0)}), \bar{\boldsymbol{g}}_2(\mathbf{W}^{(k)})\rangle \geq -\frac{1}{2}\|\bar{\boldsymbol{g}}_1(\mathbf{W}^{(k)}) - \bar{\boldsymbol{g}}_1(\mathbf{W}^{(0)})\|^2 - \frac{1}{2}\|\bar{\boldsymbol{g}}_2(\mathbf{W}^{(k)})\|^2, \tag{15}$$

and similarly:

$$\langle\bar{\boldsymbol{g}}_1(\mathbf{W}^{(0)}), \bar{\boldsymbol{g}}_2(\mathbf{W}^{(k)}) - \bar{\boldsymbol{g}}_2(\mathbf{W}^{(0)})\rangle \geq -\frac{1}{2}\|\bar{\boldsymbol{g}}_2(\mathbf{W}^{(k)}) - \bar{\boldsymbol{g}}_2(\mathbf{W}^{(0)})\|^2 - \frac{1}{2}\|\bar{\boldsymbol{g}}_1(\mathbf{W}^{(0)})\|^2. \tag{16}$$

Combining Eq.(13), Eq.(15) and Eq.(16) gives a lower bound on $\langle\bar{\boldsymbol{g}}_1(\mathbf{W}^{(k)}), \bar{\boldsymbol{g}}_2(\mathbf{W}^{(k)})\rangle$, i.e.,

$$\langle\bar{\boldsymbol{g}}_1(\mathbf{W}^{(k)}), \bar{\boldsymbol{g}}_2(\mathbf{W}^{(k)})\rangle$$
$$\geq -\frac{1}{2}\|\bar{\boldsymbol{g}}_1(\mathbf{W}^{(k)}) - \bar{\boldsymbol{g}}_1(\mathbf{W}^{(0)})\|^2 - \frac{1}{2}\|\bar{\boldsymbol{g}}_2(\mathbf{W}^{(k)})\|^2$$
$$\quad - \frac{1}{2}\|\bar{\boldsymbol{g}}_2(\mathbf{W}^{(k)}) - \bar{\boldsymbol{g}}_2(\mathbf{W}^{(0)})\|^2 - \frac{1}{2}\|\bar{\boldsymbol{g}}_1(\mathbf{W}^{(0)})\|^2 + \langle\bar{\boldsymbol{g}}_1(\mathbf{W}^{(0)}), \bar{\boldsymbol{g}}_2(\mathbf{W}^{(0)})\rangle$$
$$\geq -\frac{H^2(1 - \lambda_1^2)}{8}\|\mathbf{W}^{(k)} - \mathbf{W}^{(0)}\|^2 - \frac{1}{2}\|\bar{\boldsymbol{g}}_2(\mathbf{W}^{(k)})\|^2$$
$$\quad - \frac{H^2(1 - \lambda_1^2)}{8}\|\mathbf{W}^{(k)} - \mathbf{W}^{(0)}\|^2 - \frac{1}{2}\|\bar{\boldsymbol{g}}_1(\mathbf{W}^{(0)})\|^2 + \langle\bar{\boldsymbol{g}}_1(\mathbf{W}^{(0)}), \bar{\boldsymbol{g}}_2(\mathbf{W}^{(0)})\rangle$$

$$\geq -\frac{H^2(1-\lambda_1^2)}{4}\|\mathbf{W}^{(k)} - \mathbf{W}^{(0)}\|^2 - \frac{1}{2}\|\bar{\boldsymbol{g}}_2(\mathbf{W}^{(k)})\|^2 - \frac{1}{2}\|\bar{\boldsymbol{g}}_1(\mathbf{W}^{(0)})\|^2 + \langle\bar{\boldsymbol{g}}_1(\mathbf{W}^{(0)}), \bar{\boldsymbol{g}}_2(\mathbf{W}^{(0)})\rangle, \tag{17}$$

where the second inequality is true due to the smoothness of the loss function and:

$$\|\bar{\boldsymbol{g}}_1(\mathbf{W}^{(k)}) - \bar{\boldsymbol{g}}_1(\mathbf{W}^{(0)})\|^2 = \|\boldsymbol{g}_1(\mathbf{W}^{(k)}) - \boldsymbol{g}_1(\mathbf{W}^{(0)})\|^2 - \|\text{Proj}_{K_h^{(q)}D_1}(\boldsymbol{g}_1(\mathbf{W}^{(k)}) - \boldsymbol{g}_1(\mathbf{W}^{(0)}))\|^2$$
$$\leq (1-\lambda_1^2)\|\boldsymbol{g}_1(\mathbf{W}^{(k)}) - \boldsymbol{g}_1(\mathbf{W}^{(0)})\|^2, \tag{18}$$

as well as

$$\|\bar{\boldsymbol{g}}_2(\mathbf{W}^{(k)}) - \bar{\boldsymbol{g}}_2(\mathbf{W}^{(0)})\|^2 \leq (1-\lambda_1^2)\|\boldsymbol{g}_2(\mathbf{W}^{(k)}) - \boldsymbol{g}_2(\mathbf{W}^{(0)})\|^2. \tag{19}$$

Based on the local orthogonal-projection-based update, it can be seen that:

$$\mathbf{W}^{(k)} = \mathbf{W}^{(0)} - \alpha \sum_{i=0}^{k-1} \bar{\boldsymbol{g}}_2(\mathbf{W}^{(i)}). \tag{20}$$

Therefore, continuing with Eq.(8), we have:

$$\mathcal{F}(\mathbf{W}^{(k+1)})$$
$$\leq \mathcal{F}(\mathbf{W}^{(k)}) - [\alpha - \frac{\alpha^2 H}{2}]\|\bar{\boldsymbol{g}}_2(\mathbf{W}^{(k)})\|^2 - \alpha\langle\bar{\boldsymbol{g}}_1(\mathbf{W}^{(k)}), \bar{\boldsymbol{g}}_2(\mathbf{W}^{(k)})\rangle$$
$$\leq \mathcal{F}(\mathbf{W}^{(k)}) - [\frac{\alpha}{2} - \frac{\alpha^2 H}{2}]\|\bar{\boldsymbol{g}}_2(\mathbf{W}^{(k)})\|^2 + \frac{\alpha^3 H^2(1-\lambda_1^2)}{4}\|\sum_{i=0}^{k-1}\bar{\boldsymbol{g}}_2(\mathbf{W}^{(i)})\|^2 + \frac{\alpha}{2}\|\bar{\boldsymbol{g}}_1(\mathbf{W}^{(0)})\|^2$$
$$- \alpha\|\bar{\boldsymbol{g}}_1(\mathbf{W}^{(0)})\|\|\bar{\boldsymbol{g}}_2(\mathbf{W}^{(0)})\|, \tag{21}$$

where the last term is based on the definition of projection. Since

$$\alpha \leq \frac{\gamma\|\bar{\boldsymbol{g}}_1(\mathbf{W}^{(0)})\|}{HBK} \leq \frac{\gamma\|\bar{\boldsymbol{g}}_1(\mathbf{W}^{(0)})\|}{H\|\sum_{i=0}^{k-1}\bar{\boldsymbol{g}}_2(\mathbf{W}^{(i)})\|}, \tag{22}$$

thus

$$\frac{1}{2}\|\bar{\boldsymbol{g}}_1(\mathbf{W}^{(0)})\|^2 + \frac{\alpha^2 H^2(1-\lambda_1^2)}{4}\|\sum_{i=0}^{k-1}\bar{\boldsymbol{g}}_2(\mathbf{W}^{(i)})\|^2$$
$$\leq \frac{1}{2}\|\bar{\boldsymbol{g}}_1(\mathbf{W}^{(0)})\|^2 + \frac{\gamma^2(1-\lambda_1^2)\|\bar{\boldsymbol{g}}_1(\mathbf{W}^{(0)})\|^2}{4H^2\|\sum_{i=0}^{k-1}\bar{\boldsymbol{g}}_2(\mathbf{W}^{(i)})\|^2}H^2\|\sum_{i=0}^{k-1}\bar{\boldsymbol{g}}_2(\mathbf{W}^{(i)})\|^2$$
$$= \frac{2+\gamma^2(1-\lambda_1^2)}{4}\|\bar{\boldsymbol{g}}_1(\mathbf{W}^{(0)})\|^2. \tag{23}$$

Therefore, we can obtain that:

$$\mathcal{F}(\mathbf{W}^{(k+1)})$$
$$\leq \mathcal{F}(\mathbf{W}^{(k)}) - [\frac{\alpha}{2} - \frac{\alpha^2 H}{2}]\|\bar{\boldsymbol{g}}_2(\mathbf{W}^{(k)})\|^2 + \frac{\alpha[2+\gamma^2(1-\lambda_1^2)]}{4}\|\bar{\boldsymbol{g}}_1(\mathbf{W}^{(0)})\|^2 - \alpha\|\bar{\boldsymbol{g}}_1(\mathbf{W}^{(0)})\|\|\bar{\boldsymbol{g}}_2(\mathbf{W}^{(0)})\|$$
$$\leq \mathcal{F}(\mathbf{W}^{(k)}) - [\frac{\alpha}{2} - \frac{\alpha^2 H}{2}]\|\bar{\boldsymbol{g}}_2(\mathbf{W}^{(k)})\|^2$$
$$\leq \mathcal{F}(\mathbf{W}^{(k)}), \tag{24}$$

where the second inequality is true because:

$$\lambda_1 \geq \sqrt{1 - 2\frac{2\|\bar{\boldsymbol{g}}_2(\mathbf{W}^{(0)})\| - \|\bar{\boldsymbol{g}}_1(\mathbf{W}^{(0)})\|}{\gamma^2\|\bar{\boldsymbol{g}}_1(\mathbf{W}^{(0)})\|}}$$
$$\implies \frac{\alpha[2+\gamma^2(1-\lambda_1^2)]}{4}\|\bar{\boldsymbol{g}}_1(\mathbf{W}^{(0)})\|^2 - \alpha\|\bar{\boldsymbol{g}}_1(\mathbf{W}^{(0)})\|\|\bar{\boldsymbol{g}}_2(\mathbf{W}^{(0)})\| \leq 0. \tag{25}$$

This sufficient decrease of the objective function value indicates that the optimal $\mathcal{F}(\mathbf{W}^\star)$ can be obtained for convex loss functions.

(2) For a non-convex loss function $\mathcal{L}$, as $\nabla\mathcal{F}(\mathbf{W}^{(k)}) = \boldsymbol{g}_1(\mathbf{W}^{(k)}) + \boldsymbol{g}_2(\mathbf{W}^{(k)})$ we have Eq.(24):

$$
\begin{aligned}
&\mathcal{F}(\mathbf{W}^{(k+1)}) \\
\leq & \mathcal{F}(\mathbf{W}^{(k)}) - [\frac{\alpha}{2} - \frac{\alpha^2 H}{2}]\|\bar{\boldsymbol{g}}_2(\mathbf{W}^{(k)})\|^2 + \frac{\alpha[2+\gamma^2(1-\lambda_1^2)]}{4}\|\bar{\boldsymbol{g}}_1(\mathbf{W}^{(0)})\|^2 - \alpha\|\bar{\boldsymbol{g}}_1(\mathbf{W}^{(0)})\|\|\bar{\boldsymbol{g}}_2(\mathbf{W}^{(0)})\| \\
& - \frac{\alpha}{2}[\|\nabla\mathcal{F}(\mathbf{W}^{(k)})\|^2 - \|\boldsymbol{g}_1(\mathbf{W}^{(k)})\|^2 - \|\boldsymbol{g}_2(\mathbf{W}^{(k)})\|^2 - 2\langle\boldsymbol{g}_1(\mathbf{W}^{(k)}), \boldsymbol{g}_2(\mathbf{W}^{(k)})\rangle] \\
\leq & \mathcal{F}(\mathbf{W}^{(k)}) - [\frac{\alpha}{2} - \frac{\alpha^2 H}{2}]\|\bar{\boldsymbol{g}}_2(\mathbf{W}^{(k)})\|^2 + \frac{\alpha[2+\gamma^2(1-\lambda_1^2)]}{4}\|\bar{\boldsymbol{g}}_1(\mathbf{W}^{(0)})\|^2 - \alpha\|\bar{\boldsymbol{g}}_1(\mathbf{W}^{(0)})\|\|\bar{\boldsymbol{g}}_2(\mathbf{W}^{(0)})\| \\
& - \frac{\alpha}{2}[\|\nabla\mathcal{F}(\mathbf{W}^{(k)})\|^2 - 2\|\boldsymbol{g}_1(\mathbf{W}^{(k)})\|^2 - 2\|\boldsymbol{g}_2(\mathbf{W}^{(k)})\|^2].
\end{aligned}
\tag{26}
$$

From Eq.(20) we have

$$
\begin{aligned}
\|\boldsymbol{g}_1(\mathbf{W}^{(k)})\|^2 &= \|\boldsymbol{g}_1(\mathbf{W}^{(k)}) - \boldsymbol{g}_1(\mathbf{W}^{(0)}) + \boldsymbol{g}_1(\mathbf{W}^{(0)})\|^2 \leq 2\|\boldsymbol{g}_1(\mathbf{W}^{(k)}) - \boldsymbol{g}_1(\mathbf{W}^{(0)})\|^2 + 2\|\boldsymbol{g}_1(\mathbf{W}^{(0)})\|^2 \\
&\leq \frac{\alpha^2 H^2}{2}\|\sum_{i=0}^{k-1}\boldsymbol{g}_2(\mathbf{W}^{(i)})\|^2 + 2\|\boldsymbol{g}_1(\mathbf{W}^{(0)})\|^2 \\
&\leq \frac{\gamma^2}{2}\|\bar{\boldsymbol{g}}_1(\mathbf{W}^{(0)})\|^2 + 2\|\boldsymbol{g}_1(\mathbf{W}^{(0)})\|^2,
\end{aligned}
\tag{27}
$$

and

$$
\begin{aligned}
\|\boldsymbol{g}_2(\mathbf{W}^{(k)})\|^2 &= \|\boldsymbol{g}_2(\mathbf{W}^{(k)}) - \boldsymbol{g}_2(\mathbf{W}^{(0)}) + \boldsymbol{g}_2(\mathbf{W}^{(0)})\|^2 \leq 2\|\boldsymbol{g}_2(\mathbf{W}^{(k)}) - \boldsymbol{g}_2(\mathbf{W}^{(0)})\|^2 + 2\|\boldsymbol{g}_2(\mathbf{W}^{(0)})\|^2 \\
&\leq \frac{\alpha^2 H^2}{2}\|\sum_{i=0}^{k-1}\boldsymbol{g}_2(\mathbf{W}^{(i)})\|^2 + 2\|\boldsymbol{g}_2(\mathbf{W}^{(0)})\|^2 \\
&\leq \frac{\gamma^2}{2}\|\bar{\boldsymbol{g}}_1(\mathbf{W}^{(0)})\|^2 + 2\|\boldsymbol{g}_2(\mathbf{W}^{(0)})\|^2,
\end{aligned}
\tag{28}
$$

where the last inequality holds as

$$
\alpha \leq \frac{\gamma\|\bar{\boldsymbol{g}}_1(\mathbf{W}^{(0)})\|}{HBK} \leq \frac{\gamma\|\bar{\boldsymbol{g}}_1(\mathbf{W}^{(0)})\|}{H\|\sum_{i=0}^{k-1}\boldsymbol{g}_2(\mathbf{W}^{(i)})\|}
\tag{29}
$$

Therefore

$$
\begin{aligned}
&\mathcal{F}(\mathbf{W}^{(k+1)}) \\
\leq & \mathcal{F}(\mathbf{W}^{(k)}) - [\frac{\alpha}{2} - \frac{\alpha^2 H}{2}]\|\bar{\boldsymbol{g}}_2(\mathbf{W}^{(k)})\|^2 + \frac{\alpha[2+\gamma^2(1-\lambda_1^2)]}{4}\|\bar{\boldsymbol{g}}_1(\mathbf{W}^{(0)})\|^2 - \alpha\|\bar{\boldsymbol{g}}_1(\mathbf{W}^{(0)})\|\|\bar{\boldsymbol{g}}_2(\mathbf{W}^{(0)})\| \\
& - \frac{\alpha}{2}\|\nabla\mathcal{F}(\mathbf{W}^{(k)})\|^2 + 2\alpha\|\boldsymbol{g}_1(\mathbf{W}^{(0)})\|^2 + 2\alpha\|\boldsymbol{g}_2(\mathbf{W}^{(0)})\|^2 + \alpha\gamma^2\|\bar{\boldsymbol{g}}_1(\mathbf{W}^{(0)}\|^2 \\
\leq & \mathcal{F}(\mathbf{W}^{(k)}) - [\frac{\alpha}{2} - \frac{\alpha^2 H}{2}]\|\bar{\boldsymbol{g}}_2(\mathbf{W}^{(k)})\|^2 + \frac{\alpha[2+\gamma^2(5-\lambda_1^2)]}{4}\|\bar{\boldsymbol{g}}_1(\mathbf{W}^{(0)})\|^2 - \alpha\|\bar{\boldsymbol{g}}_1(\mathbf{W}^{(0)})\|\|\bar{\boldsymbol{g}}_2(\mathbf{W}^{(0)})\| \\
& - \frac{\alpha}{2}\|\nabla\mathcal{F}(\mathbf{W}^{(k)})\|^2 + 2\alpha\|\boldsymbol{g}_1(\mathbf{W}^{(0)})\|^2 + 2\alpha\|\boldsymbol{g}_2(\mathbf{W}^{(0)})\|^2.
\end{aligned}
\tag{30}
$$

Thus,

$$
\begin{aligned}
&\min_k \|\nabla\mathcal{F}(\mathbf{W}^{(k)})\|^2 \\
\leq & \frac{1}{K}\sum_{k=0}^{K-1}\|\nabla\mathcal{F}(\mathbf{W}^{(k)})\|^2 \\
\leq & \frac{2}{\alpha K}\sum_{k=0}^{K-1}[\mathcal{F}(\mathbf{W}^{(k)}) - \mathcal{F}(\mathbf{W}^{(k+1)})] + \frac{[2+\gamma^2(5-\lambda_1^2)]}{2(K-1)}\sum_{k=1}^{K-1}\|\bar{\boldsymbol{g}}_1(\mathbf{W}^{(0)})\|^2 - 2\|\bar{\boldsymbol{g}}_1(\mathbf{W}^{(0)})\|\|\bar{\boldsymbol{g}}_2(\mathbf{W}^{(0)})\|
\end{aligned}
$$

$$- \frac{1 - \alpha H}{K} \sum_{k=0}^{K-1} \|\bar{g}_2(\mathbf{W}^{(k)})\|^2 + 4\|g_1(\mathbf{W}^{(0)})\|^2 + 4\|g_2(\mathbf{W}^{(0)})\|^2$$

$$\leq \frac{2}{\alpha K}[\mathcal{F}(\mathbf{W}^{(0)}) - \mathcal{F}(\mathbf{W}^\star)] + \frac{[2 + \gamma^2(5 - \lambda_1^2)]}{2}\|\bar{g}_1(\mathbf{W}^{(0)})\|^2 + 4\|g_1(\mathbf{W}^{(0)})\|^2 + 4\|g_2(\mathbf{W}^{(0)})\|^2, \tag{31}$$

where the last inequality holds due to $\mathcal{F}(\mathbf{W}^\star) \leq \mathcal{F}(\mathbf{W}^{(K)})$. $\qquad\square$

## B  PROOF OF THEOREM 2

*Proof.* (1) For local orthogonal-projection-based update, we have

$$\mathbf{W}^s = \mathbf{W} - \alpha[g_2(\mathbf{W}) - \mathrm{Proj}_{K_h^{(q)}D_1}(g_2(\mathbf{W}))] = \mathbf{W} - \alpha\bar{g}_2(\mathbf{W}). \tag{32}$$

For global orthogonal-projection-based update, we have

$$\mathbf{W}^c = \mathbf{W} - \alpha[g_2(\mathbf{W}) - \mathrm{Proj}_{D_1}(g_2(\mathbf{W}))] = \mathbf{W} - \alpha\ddot{g}_2(\mathbf{W}). \tag{33}$$

Based on Eq.(8) and the smoothness of the objective function, we have an upper bound on $\mathcal{F}(\mathbf{W}^s)$:

$$\mathcal{F}(\mathbf{W}^s) \leq \mathcal{F}(\mathbf{W}) - [\alpha - \frac{\alpha^2 H}{2}]\|\bar{g}_2(\mathbf{W})\|^2 - \alpha\langle\bar{g}_1(\mathbf{W}), \bar{g}_2(\mathbf{W})\rangle, \tag{34}$$

and a lower bound on $\mathcal{F}(\mathbf{W}^c)$:

$$\mathcal{F}(\mathbf{W}^c) \geq \mathcal{F}(\mathbf{W}) + \nabla\mathcal{F}(\mathbf{W})^\top(\mathbf{W}^c - \mathbf{W}) - \frac{H}{2}\|\mathbf{W}^c - \mathbf{W}\|^2. \tag{35}$$

Combining Eq.(34) and Eq.(35), we have

$$\mathcal{F}(\mathbf{W}^s)$$

$$\leq \mathcal{F}(\mathbf{W}^c) - \nabla\mathcal{F}(\mathbf{W})^\top(\mathbf{W}^c - \mathbf{W}) + \frac{H}{2}\|\mathbf{W}^c - \mathbf{W}\|^2 - [\alpha - \frac{\alpha^2 H}{2}]\|\bar{g}_2(\mathbf{W})\|^2 - \alpha\langle\bar{g}_1(\mathbf{W}), \bar{g}_2(\mathbf{W})\rangle$$

$$= \mathcal{F}(\mathbf{W}^c) - \langle g_1(\mathbf{W}) + g_2(\mathbf{W}), -\alpha\ddot{g}_2(\mathbf{W})\rangle + \frac{\alpha^2 H}{2}\|\ddot{g}_2(\mathbf{W})\|^2 - [\alpha - \frac{\alpha^2 H}{2}]\|\bar{g}_2(\mathbf{W})\|^2$$

$$\quad - \alpha\langle\bar{g}_1(\mathbf{W}), \bar{g}_2(\mathbf{W})\rangle$$

$$= \mathcal{F}(\mathbf{W}^c) + \alpha\langle g_1(\mathbf{W}), \alpha\ddot{g}_2(\mathbf{W})\rangle + \alpha\langle g_2(\mathbf{W}), \ddot{g}_2(\mathbf{W})\rangle + \frac{\alpha^2 H}{2}\|\ddot{g}_2(\mathbf{W})\|^2 - [\alpha - \frac{\alpha^2 H}{2}]\|\bar{g}_2(\mathbf{W})\|^2$$

$$\quad - \alpha\langle\bar{g}_1(\mathbf{W}), \bar{g}_2(\mathbf{W})\rangle$$

$$= \mathcal{F}(\mathbf{W}^c) + [\alpha + \frac{\alpha^2 H}{2}]\|\ddot{g}_2(\mathbf{W})\|^2 - [\alpha - \frac{\alpha^2 H}{2}]\|\bar{g}_2(\mathbf{W})\|^2 - \alpha\langle\bar{g}_1(\mathbf{W}), \bar{g}_2(\mathbf{W})\rangle, \tag{36}$$

where the last equality is true because

$$\langle g_2(\mathbf{W}), \ddot{g}_2(\mathbf{W})\rangle = \langle\mathrm{Proj}_{D_1}(g_2(\mathbf{W})), \ddot{g}_2(\mathbf{W})\rangle + \langle\ddot{g}_2(\mathbf{W}), \ddot{g}_2(\mathbf{W})\rangle, \tag{37}$$

and both $g_1(\mathbf{W})$ and $\mathrm{Proj}_{D_1}(g_2(\mathbf{W}))$ are orthogonal to $\ddot{g}_2(\mathbf{W})$. Based on Eq.(17), the last term has:

$$\langle\bar{g}_1(\mathbf{W}), \bar{g}_2(\mathbf{W})\rangle$$

$$\geq -\frac{H^2(1 - \lambda_1^2)}{4}\|\mathbf{W} - \mathbf{W}^{(0)}\|^2 - \frac{1}{2}\|\bar{g}_2(\mathbf{W})\|^2 - \frac{1}{2}\|\bar{g}_1(\mathbf{W}^{(0)})\|^2 + \langle\bar{g}_1(\mathbf{W}^{(0)}), \bar{g}_2(\mathbf{W}^{(0)})\rangle. \tag{38}$$

Suppose that $\mathbf{W}$ is the model update at $n$-th iteration where $n \leq K$. For the local orthogonal-projection-based update,

$$\|\mathbf{W}^{(k)} - \mathbf{W}^{(0)}\|^2 = \alpha^2 \|\sum_{i=0}^{n} \bar{g}_2(\mathbf{W}^{(i)})\|^2$$

$$\leq \frac{\gamma^2 \|\bar{\boldsymbol{g}}_1(\mathbf{W}^{(0)})\|}{H^2 B^2 K^2} n \sum_{i=0}^{n} \|\bar{\boldsymbol{g}}_2(\mathbf{W}^{(i)})\|^2$$

$$\leq \frac{\gamma^2 n^2 \|\bar{\boldsymbol{g}}_1(\mathbf{W}^{(0)})\|^2}{H^2 K^2}$$

$$\leq \frac{\gamma^2 \|\bar{\boldsymbol{g}}_1(\mathbf{W}^{(0)})\|^2}{H^2}, \tag{39}$$

and similarly for global orthogonal-projection-based update, we also have

$$\|\mathbf{W}^{(k)} - \mathbf{W}^{(0)}\|^2 \leq \frac{\gamma^2 \|\bar{\boldsymbol{g}}_1(\mathbf{W}^{(0)})\|^2}{H^2}. \tag{40}$$

Therefore, continuing with Eq.(38), we obtain:

$$\langle \bar{\boldsymbol{g}}_1(\mathbf{W}^{(k)}), \bar{\boldsymbol{g}}_2(\mathbf{W}^{(k)}) \rangle$$
$$\geq -\frac{2 + \gamma^2(1 - \lambda_1^2)}{4} \|\bar{\boldsymbol{g}}_1(\mathbf{W}^{(0)})\|^2 + \|\bar{\boldsymbol{g}}_1(\mathbf{W}^{(0)})\| \|\bar{\boldsymbol{g}}_2(\mathbf{W}^{(0)})\| - \frac{1}{2} \|\bar{\boldsymbol{g}}_2(\mathbf{W})\|^2$$
$$\geq -\frac{1}{2} \|\bar{\boldsymbol{g}}_2(\mathbf{W})\|^2, \tag{41}$$

where the last inequality holds due to Eq.(25). Continuing with Eq.(36), we get:

$$\mathcal{F}(\mathbf{W}^s) \leq \mathcal{F}(\mathbf{W}^c) + [\alpha + \frac{\alpha^2 H}{2}] \|\ddot{\boldsymbol{g}}_2(\mathbf{W})\|^2 - [\frac{\alpha}{2} - \frac{\alpha^2 H}{2}] \|\bar{\boldsymbol{g}}_2(\mathbf{W})\|^2. \tag{42}$$

Based on assumption, we have

$$\|\text{Proj}_{K_h^{(q)} D_1}(\boldsymbol{g}_2(\mathbf{W}))\|_2 = \lambda_3 \|\text{Proj}_{D_1}(\boldsymbol{g}_2(\mathbf{W}))\|_2 \leq \|\text{Proj}_{D_1}(\boldsymbol{g}_2(\mathbf{W}))\|_2, \tag{43}$$

thus

$$\|\bar{\boldsymbol{g}}_2(\mathbf{W})\|^2 = \|\ddot{\boldsymbol{g}}_2(\mathbf{W})\|^2 + \|\text{Proj}_{D_1}(\boldsymbol{g}_2(\mathbf{W}))\|^2 - \|\text{Proj}_{K_h^{(q)} D_1}(\boldsymbol{g}_2(\mathbf{W}))\|^2$$
$$= \|\ddot{\boldsymbol{g}}_2(\mathbf{W})\|^2 + (1 - \lambda_3^2) \|\text{Proj}_{D_1}(\boldsymbol{g}_2(\mathbf{W}))\|^2. \tag{44}$$

Combining Eq.(42) and Eq.(44) on $\|\ddot{\boldsymbol{g}}_2(\mathbf{W})\|^2$, we have

$$\mathcal{F}(\mathbf{W}^s) \leq \mathcal{F}(\mathbf{W}^c) + [(\alpha + \frac{\alpha^2 H}{2}) - (\frac{\alpha}{2} - \frac{\alpha^2 H}{2})] \|\bar{\boldsymbol{g}}_2(\mathbf{W})\|^2 - (1 - \lambda_3^2)[\alpha + \frac{\alpha^2 H}{2}] \|\text{Proj}_{D_1}(\boldsymbol{g}_2(\mathbf{W}))\|^2$$

$$\leq \mathcal{F}(\mathbf{W}^c) + [(\frac{\alpha}{2} + \alpha^2 H)(1 - \lambda^2)] \|\boldsymbol{g}_2(\mathbf{W})\|^2 - (1 - \lambda_3^2)[\alpha + \frac{\alpha^2 H}{2}] \lambda_1'^2 \|\boldsymbol{g}_2(\mathbf{W})\|^2, \tag{45}$$

where the last inequality holds with definition 1 that $\|\text{Proj}_{D_1}(\boldsymbol{g}_2(\mathbf{W}))\| \geq \lambda_1' \|\boldsymbol{g}_2(\mathbf{W})\|$ and

$$\|\boldsymbol{g}_2(\mathbf{W})\|^2 = \|\text{Proj}_{K_h^{(q)} D_1}(\boldsymbol{g}_2(\mathbf{W})) + \bar{\boldsymbol{g}}_2(\mathbf{W})\|^2$$
$$= \|\text{Proj}_{K_h^{(q)} D_1}(\boldsymbol{g}_2(\mathbf{W}))\|^2 + \|\bar{\boldsymbol{g}}_2(\mathbf{W})\|^2$$
$$\geq \lambda_1^2 \|\boldsymbol{g}_2(\mathbf{W})\|^2 + \|\bar{\boldsymbol{g}}_2(\mathbf{W})\|^2. \tag{46}$$

Considering

$$\lambda_1 \geq \sqrt{1 - \frac{(1 - \lambda_3^2)(2 + \alpha H)\lambda_1'^2}{1 + 2\alpha H}}$$
$$\implies \alpha(1 - \lambda_1^2)(1 + 2\alpha H) \leq \alpha(1 - \lambda_3^2)(2 + \alpha H)\lambda_1'^2, \tag{47}$$

we get $\mathcal{F}(\mathbf{W}^s) \leq \mathcal{F}(\mathbf{W}^c)$.

(2) Base on the smoothness of the loss function, we have

$$\mathcal{L}_1(\mathbf{W}^{(k)}) \leq \mathcal{L}_1(\mathbf{W}^{(0)}) + \langle \boldsymbol{g}_1(\mathbf{W}^{(0)}), \mathbf{W}^{(k)} - \mathbf{W}^{(0)} \rangle + \frac{H}{4} \|\mathbf{W}^{(k)} - \mathbf{W}^{(0)}\|^2$$

$$= \mathcal{L}_1(\mathbf{W}^{(0)}) + \langle \boldsymbol{g}_1(\mathbf{W}^{(0)}), -\alpha \sum_{i=0}^{k-1} \bar{\boldsymbol{g}}_2(\mathbf{W}^{(i)}) \rangle + \frac{\alpha^2 H}{4} \| \sum_{i=0}^{k-1} \bar{\boldsymbol{g}}_2(\mathbf{W}^{(i)}) \|^2$$

$$= \mathcal{L}_1(\mathbf{W}^{(0)}) - \alpha \sum_{i=0}^{k-1} \langle \bar{\boldsymbol{g}}_1(\mathbf{W}^{(0)}), \bar{\boldsymbol{g}}_2(\mathbf{W}^{(i)}) \rangle + \frac{\alpha^2 H}{4} \| \sum_{i=0}^{k-1} \bar{\boldsymbol{g}}_2(\mathbf{W}^{(i)}) \|^2$$

$$\leq \mathcal{L}_1(\mathbf{W}^{(0)}) - \alpha \| \bar{\boldsymbol{g}}_1(\mathbf{W}^{(0)}) \| [\sum_{i=0}^{k-1} \| \bar{\boldsymbol{g}}_2(\mathbf{W}^{(i)}) \| ] + \frac{\alpha^2 H k}{4} \sum_{i=0}^{k-1} \| \bar{\boldsymbol{g}}_2(\mathbf{W}^{(i)}) \|^2. \quad (48)$$

Since $\alpha \leq \frac{4 \| \bar{\boldsymbol{g}}_1(\mathbf{W}^{(0)}) \|}{H B k^{1.5}}$, we have

$$\frac{\alpha H k}{4} \sum_{i=0}^{k-1} \| \bar{\boldsymbol{g}}_2(\mathbf{W}^{(i)}) \|^2 \leq \frac{\| \bar{\boldsymbol{g}}_1(\mathbf{W}^{(0)}) \|}{B\sqrt{k}} \sum_{i=0}^{k-1} \| \bar{\boldsymbol{g}}_2(\mathbf{W}^{(i)}) \|^2$$

$$\leq \frac{\| \bar{\boldsymbol{g}}_1(\mathbf{W}^{(0)}) \| (\sum_{i=0}^{k-1} \| \bar{\boldsymbol{g}}_2(\mathbf{W}^{(i)}) \|^2)}{\sqrt{\sum_{i=0}^{k-1} \| \bar{\boldsymbol{g}}_2(\mathbf{W}^{(i)}) \|^2}}$$

$$\leq \| \bar{\boldsymbol{g}}_1(\mathbf{W}^{(0)}) \| \sqrt{\sum_{i=0}^{k-1} \| \bar{\boldsymbol{g}}_2(\mathbf{W}^{(i)}) \|^2}$$

$$\leq \bar{\boldsymbol{g}}_1(\mathbf{W}^{(0)}) \| [\sum_{i=0}^{k-1} \| \bar{\boldsymbol{g}}_2(\mathbf{W}^{(i)}) \| ]. \quad (49)$$

Therefore, $\mathcal{L}_1(\mathbf{W}^{(k)}) \leq \mathcal{L}_1(\mathbf{W}^{(0)})$ $\qquad \square$

## C  POPULAR KERNEL FUNCTIONS

We list the popular kernel functions in Table 2. The distance $d$ can be computed by some standard distance measures such as $\ell_2$ or cosine similarity. For example, for a global representation matrix $\mathbf{R}_j^l = [\mathbf{r}_{j,1}^l, ... \mathbf{r}_{j,N^j}^l] \in \mathbb{R}^{M \times N^j}$ for layer $l$ task $j$, the distance between $a$ and $b$ on space $[N^j]$ is $d(a,b) = \arccos(\frac{\langle \mathbf{r}_{j,a}^l, \mathbf{r}_{j,b}^l \rangle}{\| \mathbf{r}_{j,a}^l \| \cdot \| \mathbf{r}_{j,b}^l \|})$, where $\mathbf{r}_{j,a}^l, \mathbf{r}_{j,b}^l$ are the $a$-th and $b$-th rows of the matrix $\mathbf{R}_j^l$.

Table 2: Popular kernel functions and their efficiencies relative to Epanechnikov kernel.

| Kernel Type | Kernel Function | Efficiency(%) |
|---|---|---|
| Uniform | $K_h(s_1, s_2) \propto \mathbf{1}[d(s_1, s_2) < h]$ | 92.9 |
| Logistic | $K_h(s_1, s_2) \propto \frac{1}{\exp(d(s_1,s_2)/h) + 2 + \exp(-d(s_1,s_2)/h)}$ | 88.7 |
| Gaussian | $K_h(s_1, s_2) \propto \frac{1}{\sqrt{2\pi}} \exp(-\frac{1}{2} h^{-2} d(s_1, s_2)^2)$ | 95.1 |
| Triangular | $K_h(s_1, s_2) \propto (1 - d(s_1,s_2)/h) \mathbf{1}[d(s_1, s_2) < h]$ | 98.6 |
| Cosine | $K_h(s_1, s_2) \propto \frac{\pi}{4} \cos \frac{\pi d(s_1,s_2)}{2h} \mathbf{1}[d(s_1, s_2) < h]$ | 99.9 |
| Epanechnikov | $K_h(s_1, s_2) \propto \frac{3}{4}[1 - (d(s_1,s_2)/h)^2] \mathbf{1}[d(s_1, s_2) < h]$ | 100 |
| Silverman | $K_h(s_1, s_2) \propto \frac{1}{2} \exp(-\frac{|d(s_1,s_2)/h|}{\sqrt{2}}) \cdot \sin(\frac{|d(s_1,s_2)/h|}{\sqrt{2}} + \frac{\pi}{4})$ | N/A |

## D  DATASETS INFORMATION

We evaluate the performance of our LMSP on four public datasets for CL: (1) Permuted MNIST (Le-Cun et al., 2010): (PMNIST) is a variant of the MNIST dataset LeCun et al. (2010), where the input pixels are randomly permuted. Following (Lopez-Paz & Ranzato, 2017; Saha et al., 2021), the dataset is divided into 10 tasks by different permutations and each task contains 10 classes; (2) CIFAR-100 Split (Krizhevsky et al., 2009): the CIFAR-100 dataset (Krizhevsky et al., 2009) is divided into 10 different tasks, and each task is a 10-way multi-class classification problem; (3)

5-Datasets (Lin et al., 2022a;b): we follow the setting of (Lin et al., 2022a;b) to use a sequence of 5 datasets, which are CIFAR-10, MNIST, SVHN (Netzer et al., 2011), not-MNIST (Bulatov, 2011), Fashion MNIST (Xiao et al., 2017), and the classification problem on each dataset is an individual task; and (4) MiniImageNet (Vinyals et al., 2016): the MiniImageNet dataset (Vinyals et al., 2016) is divided into 20 tasks, and each task includes 5 classes.

## E    ABLATION STUDIES ON KERNEL TYPE

Figure 2 shows the influence of different kernels. We adopted five different kernels in our model and the result shows that the Gaussian kernel reach the best performance. Beside, the kernel effect is not that obvious and the overall performance are similar thus we could choose the simplest one in practise to reduce the computation.

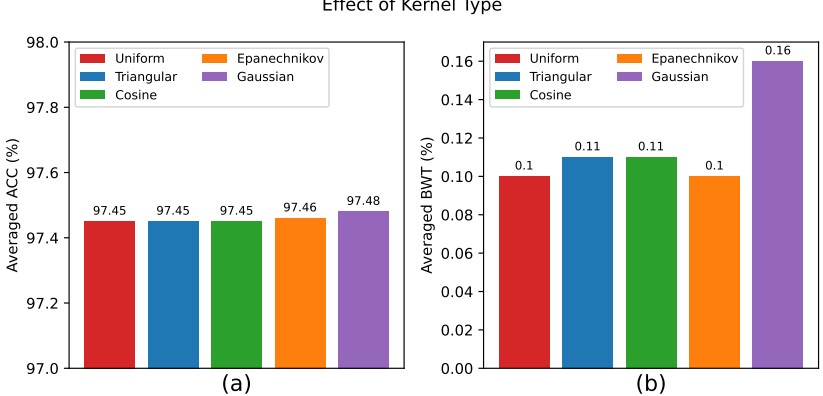

Figure 2: Ablation studies on kernel type.

## F    RESULTS OF FORWARD KNOWLEDGE TRANSFER .

We show the results of forward knowledge transfer(FWT) in the Table 3. We compared the FWT performance of our LMSP approach to those of the GPM, TRGP, and CUBER methods, which are the most related work to our paper. The value for GPM is zero because we treat GPM as the baseline and consider the relative FWT improvement over GPM. We compare them using four public datasets. We can see from the table that the FWT performance of our LMSP approach beats those of the TRGP and CUBER (two most related and state-of-the-art methods) on the PMNIST, Cifar-100 Split, and 5-Dataset datasets, and is comparable to those of the TRGP and CUBER on the MiniImageNet dataset. Clearly, this shows that the good BWT performance of our LMSP method is not achieved at the cost of sacrificing the FWT performance.

Table 3: Comparison of FWT among GPM, TRGP, CUBER and LMSP. The value for GPM is zero because we treat GPM as the baseline and consider the relative FWT improvement over GPM.

| FWT (%) | PMNIST | Cifar-100 Split | 5-Dataset | MiniImageNet |
|---|---|---|---|---|
| GPM | 0 | 0 | 0 | 0 |
| TRPG | 0.18 | 2.01 | 1.98 | 2.36 |
| CUBER | 0.80 | 2.79 | 1.96 | **3.13** |
| **LMSP(ours)** | **0.92** | **2.89** | **2.43** | 2.79 |

## G    RESULTS OF TRAINING TIME.

We show the results of forward knowledge transfer(FWT) in Table 4. As shown in the table, we summarize the normalized wall-clock training times of our LMSP algorithm and several baselines with respect to the wall-clock training time of GPM (additional wall-clock training time results can

also be found in (Saha et al., 2021)). Here, we set the rank $r$ to 5 for each local model. We can see that the wall-clock time of our LMSP method with *only one anchor point* can already reduce the total wall-clock training time of CUBER by 86% on average. Moreover, thanks to the fact that our LMSP approach endows distributed implementation that can run different local models in a parallel fashion, the total walk-clock training time with $m$ anchor points is similar to the single-anchor-point case above.

Table 4: Training time comparison on CIFAR-100 Split, 5-Datasets and MiniImageNet. Here the training time is normalized with respect to the value of GPM. Please refer (Saha et al., 2021) for more specific time.

| Training time | OWM | EWC | HAT | A-GEM | ER-Res | GPM | TRPG | CUBER | LMSP |
|---|---|---|---|---|---|---|---|---|---|
| Cifar-100 Split | 2.41 | 1.76 | 1.62 | 3.48 | 1.49 | 1 | 1.65 | 1.86 | **0.24** |
| 5-Dataset | - | 1.52 | 1.47 | 2.41 | 1.40 | 1 | 1.21 | 1.55 | **0.42** |
| MiniImageNet | - | 1.22 | 0.91 | 1.79 | 0.82 | 1 | 1.34 | 1.61 | **0.18** |

