# OpenReview forum: "Divide and Orthogonalize: Efficient Continual Learning with Local Model Space Projection"
_ICLR.cc/2024/Conference — Submitted to ICLR 2024_

### Official Review · Reviewer_apk6 · 2023-10-19

**Soundness:** 2 fair
**Presentation:** 2 fair
**Contribution:** 2 fair
**Rating:** 5
**Confidence:** 4

**Summary:**

This paper improves the efficiency of SVD decomposition in gradient-projection-based continual learning method. They introduce local model space projection (LMSP) to improve the running efficiency of SVD decomposition. At the same time, LMSP can facilitate both forward and backward transfer of gradient-projection-based methods in continual learning. The authors also provide some theoretical analysis of LMSP. Experiments on several datasets evaluate the effectiveness of the proposed method.

**Strengths:**

This paper introduces local model space projection to GPM to improve its running efficiency.

**Weaknesses:**

* This paper writing needs to be further improved.  It would be better to directly state the intuitive idea and its illustration. This would make the main idea clearer and easier to understand.


* The authors argue that SVD decomposition is computationally costly. This is true but it seems not an important problem in GPM since SVD decomposition only happens after finishing training each task, not every iteration. Therefore, the computation cost of SVD decomposition is minor compared to the overall training cost.


* The authors state that their method could reduce the complexity of SVD basis computation, but there is no empirical evaluation of the overall training efficiency improvement with the proposed method compared to the GPM itself.


* From the empirical results, LMSP improves the backward transfer, but the overall accuracy drops in some cases. The paper states that LMSP can improve both the forward and backward transfer, which does not support the claim.

**Questions:**

N/A

---

> ### Author Response · Authors · 2023-11-17
> **Response to Reviewer apk6 Part 1**
>
> We sincerely thank the reviewer's constructive comments and valuable insights, which help improve the quality of our work significantly. We have carefully revised our paper according to your comments and suggestions. Please see our revised submission, where we have highlighted all major changes in **Blue** color. Please also see our point-to-point responses as follows:
>
> > **Your Comment 1:** This paper writing needs to be further improved. It would be better to directly state the intuitive idea and its illustration. This would make the main idea clearer and easier to understand.
>
> **Our Response:** Thanks for your suggestions. We fully agree with the reviewer that adding more intuition discussions and the rationale behind our proposed LMSP approach will make the presentation of our key idea clearer and easier to understand. In this revision, we have added such discussions in introduction to further clarify our key idea: To use local low-rank approximation to reduce the complexity in continual learning with forward and backward knowledge transfers, while not sacrificing too much performance.
>
> > **Your Comment 2:** The authors argue that SVD decomposition is computationally costly. This is true but it seems not an important problem in GPM since SVD decomposition only happens after finishing training each task, not every iteration. Therefore, the computation cost of SVD decomposition is minor compared to the overall training cost.
>
> **Our Response:** Thanks for your comments. The reviewer is correct that one round of layer-wise SVD operations is performed in GPM after finishing the learning of a new task. In fact, all orthogonal-projection-based CL approaches (not only GPM, but also TRGP, CUBER, and our LMSP methods) all perform SVD once for each layer after the training of each task. However, we note that such a one-round layer-wise per-task SVD does **not** necessarily mean that the resultant computation is cheap. In fact, such SVD computations remain highly expensive. Specifically, note that we need to perform SVD for each layer. With the ever-increasing widths and depths of large and deep learning models, computing one SVD even just for one layer becomes more and more difficult due to the $\mathcal{O}(n^3)$ complexity as the width $n$ of each layer gets large.
>
> On the other hand, we note that the training cost of each task is **not** necessarily higher than performing SVD, as the total number of iterations of most first-order methods typically does *not* scale with the model size/dimension. In our experiments, we find that the processing time of SVD is significantly higher than those of other components of the model. This is also evidenced by our newly added walk-time comparison experiments in the table in the response to your Comment 3 below. In that table below, we summarize the wall-clock training times of our LMSP algorithm is much **shorter** than those baselines with full SVDs.
>
> All these results and analyses suggest that the computational complexity of SVD is the paint point of orthogonal-projection-based CL approaches. In addition, it is also mentioned in CUBER (Lin et al, 2022) that running a full SVD is time-consuming, which is consistent with our observations.

---

> ### Author Response · Authors · 2023-11-17
> **Response to Reviewer apk6 Part 2**
>
> > **Your Comment 3:** The authors state that their method could reduce the complexity of SVD basis computation, but there is no empirical evaluation of the overall training efficiency improvement with the proposed method compared to the GPM itself.
>
> **Our Response:** Thanks for your comments. In this rebuttal period, we have added an additional set of experiments to evaluate the wall-clock training time  of our LMSP approach and compare with several closely related baselines. In the table below, we summarize the normalized wall-clock training times of our LMSP algorithm and several baselines with respect to the wall-clock training time of GPM (additional wall-clock training time results can also be found in [R3]). Here, we set the rank $r$ to 5 for each local model. We can see that the wall-clock time of our LMSP method with *only one anchor point* can already reduce the total wall-clock training time of CUBER by 86% on average. Moreover, thanks to the fact that our LMSP approach endows distributed implementation that can run different local models in a parallel fashion, the total walk-clock training time with $m$ anchor points is similar to the single-anchor-point case above.
>
>
> | Training time | Cifar-100 Split | 5-Dataset | MiniImageNet |
> | ---------------- | --- | --- | --- |
> |OWM               | 2.41| -   | -   |
> |EWC               | 1.76| 1.52| 1.22|
> |HAT               | 1.62| 1.47| 0.91|
> |A-GEM             | 3.48| 2.41| 1.79|
> |ER_Res            | 1.49| 1.40| 0.82|
> |GPM               | 1.00| 1.00| 1.00|
> |TRPG              | 1.65| 1.21| 1.34|
> |CUBER             | 1.86| 1.55| 1.61|
> |**LMSP (Ours)**   |**0.24**|**0.42**|**0.18**|
>
> [R1] S. Gobinda et al., "Gradient Projection Memory for Continual Learning," in Proc. ICLR 2020.
>
> > **Your Comment 4:** From the empirical results, LMSP improves the backward transfer, but the overall accuracy drops in some cases. The paper states that LMSP can improve both the forward and backward transfer, which does not support the claim.
>
> **Our Response:** Thanks for your comments. We would like to clarify that, due to the information loss of using local model approximation in our LMSP method, it could happen that the overall accuracy of LMSP may be outperformed by other baseline methods. However, we want to emphasize that our goal in this paper is to significantly reduce the computational complexity from $\mathcal{O}(n^3)$ to $\mathcal{O}(n^2)$ by using local model approximation, even though this could lead to a slight performance loss. In other words, we would like to pursue *low-complexity CL algorithmic design* by potentially and slightly trading-off learning performance.
>
> Also, in this rebuttal period, we have added additional experiments to evaluate the forward knowledge transfer (FWT) performance. As shown in the following table, we compared the FWT performance of our LMSP approach to those of the GPM, TRGP, and CUBER methods, which are the most related work to our paper. The value for GPM is zero because we treat GPM as the baseline and consider the relative FWT improvement over GPM. We compare them using four public datasets. We can see from the table that the FWT performance of LMSP outperforms those of TRGP and CUBER (two most related and state-of-the-art methods) on the PMNIST, Cifar-100 Split, and 5-Dataset datasets, and is comparable to those of TRGP and CUBER on the MiniImageNet dataset. This shows that LMSP does improve both FWT and BWT in most cases.
>
> | FWT (%) | PMNIST | Cifar-100 Split | 5-Dataset | MiniImageNet |
> | ---------------- | --- | --- | --- | --- |
> |GPM               | 0 | 0 | 0 | 0 |
> |TRPG              | 0.18| 2.01| 1.98| 2.36|
> |CUBER             | 0.80| 2.79| 1.96| **3.13**|
> |**LMSP (Ours)**   | **0.92**| **2.89** | **2.43** | 2.79|

---

### Official Review · Reviewer_CGvu · 2023-10-27

**Soundness:** 2 fair
**Presentation:** 2 fair
**Contribution:** 2 fair
**Rating:** 6
**Confidence:** 3

**Summary:**

Based on the basic framework of orthogonal-projection-based CL methods, this article proposes a local model space projection (LMSP) based efficient continual learning framework to help reduce the complexity of computation. The authors provide a theoretical analysis of backward knowledge transfer. Experiments based on multiple datasets demonstrate the effectiveness of the method.

**Strengths:**

- The paper is well-structured with clear writing.
- Leveraging the problem definition from previous research, this study presents a novel local model space projection approach, optimizing continual learning.
- The authors also provide a theoretical analysis of the convergence.

**Weaknesses:**

- The problem definition, framework, and convergence analysis of this work are derived from existing work. While the efficiency approach is intuitive and easy to understand, its novelty causes me concern.
- The authors use local low-rank matrices defined by anchor points to approximate each layer parameter matrix. However, the accuracy of this approximation, and in particular how it is affected by m, is not discussed. Moreover, the proposed framework and analysis also ignore this issue.
- The author introduces LLRA to improve computational efficiency. However, they do not perform experiments to evaluate the computational complexity and specifically do not show the saved wall-clock time compared with the LRA method.

**Questions:**

- The author states that there is no significant difference between the two methods in selecting anchor points. Can you give some intuitive explanation?
- Is there some relationship between ranking and the number of anchors?

---

> ### Author Response · Authors · 2023-11-17
> **Response to Reviewer CGvu Part 1**
>
> We sincerely thank the reviewer's constructive comments and valuable insights, which help improve the quality of our work significantly. We have carefully revised our paper according to your comments and suggestions. Please see our revised submission, where we have highlighted all major changes in **Blue** color. Please also see our point-to-point responses as follows:
>
> > **Your Comment 1:** The problem definition, framework, and convergence analysis of this work are derived from existing work. While the efficiency approach is intuitive and easy to understand, its novelty causes me concern.
>
> **Our Response:** Thanks for the comments. We would like to point out that, although the continual learning (CL) problem definition and framework in this paper are not completely new, it does *not* necessarily mean that there is no more research to be done. We note that CL is a broad and very active research field in recent years (see [R1] for the latest survey). However, there remains a large number of open fundamental research problems in CL and the performance of existing CL methods are still far from satisfactory. In this paper, we note that even the state-of-the-art orthogonal-projection-based CL approaches (TRGP and CUBER) still suffer from *high computational complexity* due to their use of the expensive SVD operations. This problem is further exacerbated by the ever-increasing large and deep vision and language models in the CL regime (i.e., sequential multi-task training). Therefore, our goal in this paper is to develop **new** orthogonal-projection-based CL methods with a significantly lower computational complexity. Toward this end, we propose a local model space projection (LMSP) approach, which is **new** in the CL literature.
>
> For the convergence analysis for our LMSP method, it is true that our proof is based on the framework of first-order optimization algorithm convergence analysis, which starts from bounding one-step descent and finishes at telescoping one-step descent and rearranging to arrive at stationarity gap bound. However, we point out that the similarity of our convergence performance analysis compared to other methods ends there. The complications arising from the use of local model projection approximations renders our convergence proof significantly different from those of existing CL methods. Specifically, our convergence proof and analysis involves the new notion called "local relative orthogonality" (see Definition 5 in our revised paper). Theorem 1 focuses on proving the convergence of our local algorithm and Theorem 2 proves that under such conditions and Definition 5, the out LMSP could achieve even better results than the global algorithm counterpart (CUBER and TRGP).
>
> [R1] L. Wang et al., "A Comprehensive Survey of Continual Learning: Theory, Method and Application," https://arxiv.org/abs/2302.00487
>
> > **Your Comment 2:** The authors use local low-rank matrices defined by anchor points to approximate each layer parameter matrix. However, the accuracy of this approximation, and in particular how it is affected by $m$, is not discussed. Moreover, the proposed framework and analysis also ignore this issue.
>
> **Our Response:** Thanks for your comments. In Section 5, we have provided in-depth ablation studies on the impacts of different rank values $r$ in low-rank approximations and different number of anchor points $m$ on the learning accuracy (ACC) and backward knowledge transfer (BWT). More specifically, with the use of local model approximation to reduce computational complexity, information loss of the original learning model is inevitable. Our goal in this paper is to reduce the computational complexity without sacrificing too much performance.
>
> Also, we did not ignore the impacts of $m$ in our theoretical analysis since these local model approximation errors have already been implicitly captured in $\bar{\mathbf{g}}_i(\mathbf{W})$. As shown in the paper, the local model approximation affects the convergence analysis through the $\bar{\mathbf{g}}_i(\mathbf{W})$, $i=1,2$. Thus, by choosing the top-K correlated local tasks, the more anchor points we have, the smaller approximation error of $\bar{\mathbf{g}}_i(\mathbf{W})$ compared to their true versions $\ddot{\mathbf{g}_i}(\mathbf{W})$, $i=1,2$ we get. Moreover, with the approximation error bound from [R2], we can theoretically characterize the impact of $m$ in our analysis.
>
> [R2] J. Lee et al., "Local Low-Rank Matrix Approximation," in Proc. ICML 2013.

---

> > ### Comment · Reviewer_CGvu · 2023-11-20
> > **Response**
> >
> > 1. To clarify, I did not intend to suggest that the field of continual learning has been fully explored. I agree with Reviewer apk6 regarding the presentation of the manuscript. A concise explanation of existing frameworks and a focused discourse on the unique aspects of your method, would enhance the paper's readability and effectively highlight its novel contributions.
> >
> > 2. Regarding the relationship between rank and the number of anchor points, my understanding is as follows: The rank reflects the number of local modes within the representation matrix. The number of anchor points influences the accuracy of this local approximation. For matrices with few information, a lower rank suggests that fewer anchor points are needed to accurately represent the information. Both your experiments and response suggest that these two elements function independently.
> >
> > 3. Concerning the two methods of selecting anchor points: random selection may result in points that are too similar or possess overlapping information, whereas pre-clustering to find centroids is likely to provide a more distinct and diverse representation. I am unsure why both methods are deemed equally viable. Additionally, I'm curious about the role of data bias in relation to these selection methods.

---

> > > ### Author Response · Authors · 2023-11-22
> > > **Response to Reviewer CGvu's Response**
> > >
> > > > **Your Comment 1:** To clarify, I did not intend to suggest that the field of continual learning has been fully explored. I agree with Reviewer apk6 regarding the presentation of the manuscript. A concise explanation of existing frameworks and a focused discourse on the unique aspects of your method, would enhance the paper's readability and effectively highlight its novel contributions.
> > >
> > > **Our Response:** Thanks for your agreement that the field of continual learning still has many open and foundational problems to study. In this revision, we have already added discussions and highlighted the differences and novelty of our work in introduction with **Blue** color, please check our revised submission.
> > >
> > > > **Your Comment 2:** Regarding the relationship between rank and the number of anchor points, my understanding is as follows: The rank reflects the number of local modes within the representation matrix. The number of anchor points influences the accuracy of this local approximation. For matrices with few information, a lower rank suggests that fewer anchor points are needed to accurately represent the information. Both your experiments and response suggest that these two elements function independently.
> > >
> > > **Our Response:** Thanks for your comments. It appears that there are still some misunderstandings on the local model approximation approach. We want to emphasize that the number of anchor points $n$ and the rank $r$ are indeed two **independent** parameters. Here, we would like to first further clarify the local low-rank approximation approach used in our LMSP-based CL algorithm.
> > >
> > > To perform local low-rank approximation (cf. [Lee et al. ICML'13]), one will first decide the number of anchor points $n$ to project the original high-rank matrix into a set of local low-rank matrices, i.e., the number of anchor points defines the number of local models (for visual illustration, see Fig. 1 in [Lee et al. ICML'13] at https://jmlr.org/papers/volume17/14-301/14-301.pdf). Then, for each projected local matrix (i.e., a local model), we compute a rank-$r$ approximation, where rank $r$ is another parameter we can choose. From this procedure, it can be seen that $n$ and $r$ are two **independent** parameters to be chosen (i.e., $n$ is not necessarily determined by $r$ and vice versa). Moreover, they jointly determine the overall accuracy of the local low-rank approximation. The more anchor points $n$ in use and the higher the chosen rank $r$, the smaller the overall approximation error of local low-rank approximation. In the special case where we use full rank and set number of anchor points to be 1, then our LMSP-based CL method (i.e., with local low-rank approximation) reduces to the CUBER baseline method.
> > >
> > > As mentioned in our first response, in practice, we often prefer to choose a small rank value $r$ since it will significantly reduce the computational complexity. Also, subject to computational resource limits, choosing more anchor points is more preferable, since this would yield a better approximation to the original model. Moreover, since each local model approximation could run in a parallel fashion (implied by our LMSP method's distributed implementation), having more anchor points will not significantly increase the wall clock time performance.
> > >
> > > > **Your Comment 3:** Concerning the two methods of selecting anchor points: random selection may result in points that are too similar or possess overlapping information, whereas pre-clustering to find centroids is likely to provide a more distinct and diverse representation. I am unsure why both methods are deemed equally viable. Additionally, I'm curious about the role of data bias in relation to these selection methods.
> > >
> > > **Our Response:** Thanks for your comments. We agree that pre-clustering to find centroids is likely to provide a more distinct and diverse representation and it is also proved by some works such as [R1]. The improvements of accuracy is reported to be around 2% in MovieLens-1M. However, in the random selection approach, as long as the choices of random anchor points are relatively uniform, the empirical difference between two selection methods is not significant based on our numerical experience. Considering the additional computational costs introduced by clustering methods (e.g., k-means), such a marginal improvement (at least in the CL applications and experiments we conducted) may not justify the additional costs of doing pre-clustering. Thus, we have adopted random anchor points selection in our experiments for lower implementation complexity. That being said, we do not rule out the possibility that pre-clustering may be favorable in other applications of local low-rank approximation, but this is beyond the scope of the continual learning applications we focus on in this paper.
> > >
> > > [R1] M. Zhang et al.,"Local Low-Rank Matrix Approximation with Preference Selection of Anchor Points," in Proc. WWW 2017.

---

> > > > ### Comment · Reviewer_CGvu · 2023-11-22
> > > > **Reply**
> > > >
> > > > Thanks for your response. My comments have been addressed, so I'll bump up the score.

---

> > > > > ### Author Response · Authors · 2023-11-22
> > > > > **Thank you!**
> > > > >
> > > > > Thank you very much! We wish to express our appreciation for your in-depth comments and suggestions, which have greatly improved the manuscript.

---

> ### Author Response · Authors · 2023-11-17
> **Response to Reviewer CGvu Part 2**
>
> > **Your Comment 3:** The author introduces LLRA to improve computational efficiency. However, they do not perform experiments to evaluate the computational complexity and specifically do not show the saved wall-clock time compared with the LRA method.
>
> **Our Response:** Thanks for your comments. We note that we do have experimental results to evaluate the computational complexity of our LMSP approach. In the table below, we summarize the normalized wall-clock training times of our LMSP algorithm and several baselines with respect to the wall-clock training time of GPM (additional wall-clock training time results can also be found in [R3]). Here, we set the rank $r$ to 5 for each local model. We can see that the wall-clock time of our LMSP method with *only one anchor point* can already reduce the total wall-clock training time of CUBER by 86% on average. Moreover, thanks to the fact that our LMSP approach endows distributed implementation that can run different local models in a parallel fashion, the total walk-clock training time with $m$ anchor points is similar to the single-anchor-point case above.
>
> | Training Time | Cifar-100 Split | 5-Dataset | MiniImageNet |
> | ---------------- | --- | --- | --- |
> |OWM               | 2.41| -   | -   |
> |EWC               | 1.76| 1.52| 1.22|
> |HAT               | 1.62| 1.47| 0.91|
> |A-GEM             | 3.48| 2.41| 1.79|
> |ER_Res            | 1.49| 1.40| 0.82|
> |GPM               | 1.00| 1.00| 1.00|
> |TRPG              | 1.65| 1.21| 1.34|
> |CUBER             | 1.86| 1.55| 1.61|
> |**LMSP (Ours)**   |**0.24**|**0.42**|**0.18**|
>
> [R3] S. Gobinda et al., "Gradient Projection Memory for Continual Learning," in Proc. ICLR 2020.
>
> > **Your Comment 4:** The author states that there is no significant difference between the two methods in selecting anchor points. Can you give some intuitive explanation?
>
> **Our Response:** Thanks for the suggestion. As we discussed in the paper, if the new task is strongly correlated with some old tasks, there should be better knowledge transfer from the correlated old tasks to the new task. Thus, the performance largely relies on finding correlated tasks. Supposing the data is not biased, simply choosing enough anchor points should provide enough candidates for the new task to choose.
>
> > **Your Comment 5:** Is there some relationship between ranking and the number of anchors?
>
> **Our Response:** Thanks for your question. In theory, the values of rank and number of anchors can be chosen independently and arbitrarily in our LMSP approach. In the extreme case, if we use full rank and set number of anchor points to be 1, then our LMSP method reduces to the CUBER baseline method. In practice, we often prefer to choose a small rank value since it will significantly reduce the computational complexity. Also, if permitted by computational resources, choosing more anchor points is more preferable, since this would yields better approximation to the original model. Moreover, since each local model approximation could run in a parallel fashion (implied by our LMSP method's distributed implementation), having more anchor points will not significantly increase the wall clock time performance.

---

### Official Review · Reviewer_Shvn · 2023-10-31

**Soundness:** 3 good
**Presentation:** 4 excellent
**Contribution:** 3 good
**Rating:** 8
**Confidence:** 3

**Summary:**

This paper proposes Local Model Space Projection, a method in continual learning that aims at avoiding forgetting and encourage knowledge transfer by performing orthogonal updates of parameters over the sequence of tasks. The method considers three regimes: 1) forgetting avoidance, 2) forward transfer, 3) backward transfer, which can be represented as variants of the problem of finding orthogonal directions for parameter updates. The method constructs local model spaces of each task by selecting some anchoring points from the task's representation. Using these representations, a similarity between tasks can be measured across local representations to determine whether the task has local sufficient projection, local positive correlation or local relative orthogonality. Theoretical analyses along with experimental results are provided. Experiments are reported for 4 benchmark datasets, and compared to a range of SOTA continual learning methods from different families (regularization, replay, orthogonalization). Results are provided in terms of accuracy and backward transfer.

**Strengths:**

- The paper proposes an original method that exploits the idea of orthogonal projections to learn new tasks whilst controlling forgetting and encouraging forward and backward transfer. The consideration of particular regimes for each of these problems, and the fact that each of these regimes can be addressed with the same underlying idea of projections that consider local representations of tasks seems novel and useful.
- The paper is very clear, easy to follow and mostly complete as it considers both theoretical and experimental demonstrations of how and why it works.
- The paper is somewhat significant in the sense that it seemingly not only tackles forgetting but also knowledge transfer, and it presents some good results in both accuracy and backward transfer.

**Weaknesses:**

- Although the proposed method seems quite competitive in terms of experimental results, there is no report on the performance of forward transfer. This is extremely relevant as forward and backward transfer are usually in trade-off (the more forward, the less backward transfer and vice-versa). How can you guarantee that the good results in backward transfer do not require sacrificing forward transfer, or even just the fact of learning the new task reasonably well?

**Questions:**

- Can you provide actual performance numbers for forward transfer?

---

> ### Author Response · Authors · 2023-11-17
> **Response to Reviewer Shvn**
>
> We sincerely thank the reviewer's constructive comments and valuable insights, which help improve the quality of our work significantly. We have carefully revised our paper according to your comments and suggestions. Please see our revised submission, where we have highlighted all major changes in **Blue** color. Please also see our point-to-point responses as follows:
>
> > **Your Comment 1:** Although the proposed method seems quite competitive in terms of experimental results, there is no report on the performance of forward transfer. This is extremely relevant as forward and backward transfer are usually in trade-off (the more forward, the less backward transfer and vice-versa). How can you guarantee that the good results in backward transfer do not require sacrificing forward transfer, or even just the fact of learning the new task reasonably well?
>
> **Our Response:** Thanks for this suggestion. In this rebuttal period, we have added additional experiments to evaluate the forward knowledge transfer (FWT) performance. As shown in the following table, we compared the FWT performance of our LMSP approach to those of GPM, TRGP, and CUBER methods, which are the most related work to our paper. The value for GPM is zero because we treat GPM as the baseline and consider the relative FWT improvement over GPM. We compare them using four public datasets. We can see from the table that the FWT performance of our LMSP approach beats those of the TRGP and CUBER (two most related and state-of-the-art methods) on the PMNIST, Cifar-100 Split, and 5-Dataset datasets, and is comparable to those of the TRGP and CUBER on the MiniImageNet dataset. Clearly, this shows that the good BWT performance of our LMSP method is **not** achieved at the cost of sacrificing the FWT performance.
>
> | FWT (%) | PMNIST | Cifar-100 Split | 5-Dataset | MiniImageNet |
> | ---------------- | --- | --- | --- | --- |
> |GPM               | 0 | 0 | 0 | 0 |
> |TRPG              | 0.18| 2.01| 1.98| 2.36|
> |CUBER             | 0.80| 2.79| 1.96| **3.13**|
> |**LMSP (Ours)**   | **0.92**| **2.89** | **2.43** | 2.79|
>
> > **Your Comment 2:** Can you provide actual performance numbers for forward transfer?
>
> **Our Response:** Thanks for the suggestion. Please see the table above.

---

### Official Review · Reviewer_ZQXi · 2023-10-31

**Soundness:** 2 fair
**Presentation:** 2 fair
**Contribution:** 2 fair
**Rating:** 5
**Confidence:** 2

**Summary:**

This paper studies an interesting topic, continual learning, which aims to learn a series of tasks without forgetting. . The existing CL methods require either an extensive amount of resources for computing gradient projections or storing a large number of old tasks’ data.  . In this paper, a local model space projection
(LMSP) is proposed to not only significantly reduce the complexity of computation, but also enables forward and backwardknowledge transfer. Extensive experiments on several public datasets demonstrate the efficiency of our approach.

**Strengths:**

1. This paper is well-written.
2. The research topic is very interesting.

**Weaknesses:**

1. Performing the forward and backward knowledge transfer has been done in the existing works.
2. The proposed approach relies on the task information, which can not be used in task-free continual learning.
3. The proposed approach does not always achieve the best performance in some datasets.
4. Although the proposed approach can reduce computational costs but would increase more parameters.

**Questions:**

Please see the weakness section.

---

> ### Author Response · Authors · 2023-11-17
> **Response to Reviewer ZQXi Part 1**
>
> We sincerely thank the reviewer's constructive comments and valuable insights, which help improve the quality of our work significantly. We have carefully revised our paper according to your comments and suggestions. Please see our revised submission, where we have highlighted all major changes in **Blue** color. Please also see our point-to-point responses as follows:
>
> > **Your Comment 1:** Performing the forward and backward knowledge transfer has been done in the existing works.
>
> **Our Response:** Thanks for the comments. Although it is true that both forward knowledge transfer (FWT) and backward knowledge transfer (BWT) have been studied in the literature of continual learning (CL), the FWT and BWT performances of existing work in this area *remain far from satisfactory*. More specifically, the FWT and BWT in the existing CL methods require either an extensive amount of resources for computing gradient projections (in orthogonal-projection-based CL) or storing a large amount of old tasks’ data (in experience-replay-based and regularization-based CL). The limitations of these existing work motivate us to propose a new CL method to improve the FWT and BWT performances.
>
> Specifically, in this paper, we focus on reducing the $\mathcal{O}(n^3)$ computational complexity in SVD and also increasing the scalability of orthogonal-projection-based CL methods. This is due to the nice fact that orthogonal-projection-based CL methods do not need to access old tasks' data. Toward this end, we propose a local model space projection (LMSP) approach that could achieve $\mathcal{O}(n^2)$ instead of $\mathcal{O}(n^3)$ complexity.

---

> ### Author Response · Authors · 2023-11-17
> **Response to Reviewer ZQXi Part 2**
>
> > **Your Comment 2:** The proposed approach relies on the task information, which can not be used in task-free continual learning.
>
> **Our Response:** Thanks for your comments. We clarify that our focus in this paper is the standard task-based CL setting, i.e., tasks arrive at the learner *sequentially* with clear task boundaries (see, e.g., [R1] for the description of this standard setting of CL). However, we would like to point out that our work focuses on the *orthogonal-projection-based* CL approach, which requires the *least* (in fact, almost zero) amount of task information since orthogonal-projection-based CL methods do *not* need to save any old tasks data. All we need is to compute the new null space of the model parameters upon finishing the learning of the previous task.
>
> On the other hand, we also note that "task-free continual learning" is a new CL paradigm, which refers to CL systems with **no** clear boundaries between tasks and data distributions of tasks gradually and continuously changing (see [R2] for the detailed description of task-free CL). Clearly, task-free CL is a more complex CL paradigm. How to conduct CL without requiring previous tasks' information is a far more challenging open problem in the community, which deserves an independent paper dedicated to this topic. But this is beyond the scope of our current work, and could be an interesting and important future direction. We thank the reviewer for suggesting this direction.
>
> [R1] L. Wang et al., "A Comprehensive Survey of Continual Learning:
> Theory, Method and Application," https://arxiv.org/abs/2302.00487
>
> [R2] R. Aljundi et al., "Task-Free Continual Learning," in Proc. CVPR 2019.
>
> > **Your Comment 3:** The proposed approach does not always achieve the best performance in some datasets.
>
> **Our Response:** Thanks for your comments. We would like to clarify that, due to the information loss of using local model approximation in our LMSP method, it could happen that LMSP may be outperformed by other baseline methods. However, we want to emphasize that our goal in this paper is to significantly reduce the computational complexity from $\mathcal{O}(n^3)$ to $\mathcal{O}(n^2)$ by using local model approximation, even though this could lead to a slight performance loss. In other words, we would like to pursue *low-complexity CL algorithmic design* by potentially and slightly trading-off learning performance.
>
> Interestingly, our experiments show that, due to other complex factors in CL systems, our LMSP approach actually *outperforms* the baseline approaches in most scenarios (cf. Table 1). Also, it is worth noting that we theoretically characterized the conditions under which our LMSP approach could achieve better results.
>
> > **Your Comment 4:** Although the proposed approach can reduce computational costs but would increase more parameters.
>
> **Our Response:** Thanks for the comments. It appears that there are some misunderstandings and confusions that are perhaps due to the relatively complex math notations in our algorithm. We would like to clarify that our local model projection approach does *not* increase the number of model parameters (i.e., our LMSP approach remains having the same number of parameters compared to CUBER, which is the most related work). Specifically, we apply local model approximation on each layer's output representation by partitioning the layer's matrix into smaller submatrices (defined by anchor points), which allows faster processing of these smaller submatrices *in parallel*. During this process, the total number of parameters remains the same (cf. the description at the bottom of Page 4 and Eq. (3)). Then, our LMSP method updates the new weights $\mathbf{W}^l$ from previous model weights using the LMSP-based projected gradients and scaling parameters, hence the number of parameters remains the same as those of CUBER (cf. Eq. (7)).

---

### Meta-Review · Area_Chair_2ty3 · 2023-12-07

**Metareview:**

This paper proposes a local model space projection (LMSP) method to reduce the heavy computation cost of the SVD step in orthogonal projection-based continual learning. The authors also theoretically estimate that the complexity can be reduced to O(n^2). Furthermore, extensive experiments demonstrate the efficiency of the proposed method.

Strengths:

(1)   This paper is well written. The motivation of this work is clear, it focuses on reducing the computing time of the projection step.

(2)   The authors also try to provide computation complexity and convergence analysis for the proposed methods.

Weaknesses:

(1)   As pointed out by Reviewer apk6, SVD decomposition only happens after finishing training each task, not every iteration. The claimed speedup ratio is questionable and problem-dependent. If the number of streaming data for each epoch is very large, the training time for each task will dominate the SVD process. The authors should further clarify this point in the next version.

(2)   The provided theory mainly focuses on convergence, which cannot reflect the efficacy of the proposed approach. From the convergence results, we know that the approximation does not affect the convergence. However, the final test accuracy is actually affected according to the reported performance. It is better to analyze how the local approximal error affects the overall generalization performance, which will reflect the effectiveness of the proposed approach better.

(3)   Actually, there exist extensive variants of SVD methods, e.g., randomized PCA, Krylov method, truncated SVD, etc. In general, truncated SVG is 10X faster than vanilla SVD (check: https://dummeraugust.com/main/content/blog/posts/demos/svd_and_truncated_svd/index.php). For a fair comparison, the authors should give more experiment comparisons for the proposed methods with the popular variants of SVD methods in the continual setting, which can better reflect the necessity of the proposed new component.

In summary, after the authors' response, the evaluation of the novelty and efficacy of the proposed approach is not sufficient. This work needs more effort to demonstrate the efficacy of the proposed approach.  Therefore, I recommend the rejection.

**Justification For Why Not Higher Score:**

N/A

**Justification For Why Not Lower Score:**

N/A

---

### Decision · Program_Chairs · 2024-01-16

Reject